# Identification of 5-HT$_{2A}$ receptor signaling pathways associated with psychedelic potential

Jason Wallach[1] ✉, Andrew B. Cao[2], Maggie M. Calkins [2], Andrew J. Heim [3,9], Janelle K. Lanham [2], Emma M. Bonniwell [2], Joseph J. Hennessey[2], Hailey A. Bock [2], Emilie I. Anderson[2], Alexander M. Sherwood[4], Hamilton Morris[1], Robbin de Klein[5], Adam K. Klein [6,10], Bruna Cuccurazzu [6], James Gamrat[1], Tilka Fannana[1], Randy Zauhar[3,11], Adam L. Halberstadt[5,6,12] ✉ & John D. McCorvy [2,7,8] ✉

Serotonergic psychedelics possess considerable therapeutic potential. Although 5-HT$_{2A}$ receptor activation mediates psychedelic effects, proto-typical psychedelics activate both 5-HT$_{2A}$-Gq/11 and β-arrestin2 transducers, making their respective roles unclear. To elucidate this, we develop a series of 5-HT$_{2A}$-selective ligands with varying Gq efficacies, including β-arrestin-biased ligands. We show that 5-HT$_{2A}$-Gq but not 5-HT$_{2A}$-β-arrestin2 recruitment efficacy predicts psychedelic potential, assessed using head-twitch response (HTR) magnitude in male mice. We further show that disrupting Gq-PLC signaling attenuates the HTR and a threshold level of Gq activation is required to induce psychedelic-like effects, consistent with the fact that certain 5-HT$_{2A}$ partial agonists (e.g., lisuride) are non-psychedelic. Understanding the role of 5-HT$_{2A}$ Gq-efficacy in psychedelic-like psychopharmacology permits rational development of non-psychedelic 5-HT$_{2A}$ agonists. We also demonstrate that β-arrestin-biased 5-HT$_{2A}$ receptor agonists block psychedelic effects and induce receptor downregulation and tachyphylaxis. Overall, 5-HT$_{2A}$ receptor Gq-signaling can be fine-tuned to generate ligands distinct from classical psychedelics.

Classical (serotonergic) psychedelics have undergone a resurgence of interest for their potential to produce rapid and sustained therapeutic effects[1,2]. The therapeutic properties of psychedelics are being explored both preclinically and clinically, but studies have focused on a limited number of compounds[3,4]. Psychedelics may be limited by their hallucinogenic effects, which can cause confusion and anxiety in some patients, necessitating close clinical supervision. Recent pre-clinical work, however, suggests it may be possible to disentangle

[1]Department of Pharmaceutical Sciences, Saint Joseph's University, Philadelphia, PA 19104, USA. [2]Department of Cell Biology, Neurobiology, and Anatomy, Medical College of Wisconsin, Milwaukee, WI 53226, USA. [3]Department of Chemistry, Saint Joseph's University, Philadelphia, PA 19104, USA. [4]Usona Institute, Madison, WI 53711, USA. [5]Research Service, VA San Diego Healthcare System, San Diego, CA 92161, USA. [6]Department of Psychiatry, University of California San Diego, La Jolla, CA 92093, USA. [7]Neuroscience Research Center, Medical College of Wisconsin, Milwaukee, WI 53226, USA. [8]Cancer Center, Medical College of Wisconsin, Milwaukee, WI 53226, USA. [9]Present address: Chemical Computing Group ULC, 910-1010 Sherbrooke W, Montréal, QC H3A 2R7, Canada. [10]Present address: Gilgamesh Pharmaceuticals, New York, NY 10003, USA. [11]Present address: Artemis Discovery, LLC, Suite 300, 709 N 2nd Street, Philadelphia, PA 19123, USA. [12]Present address: Center for Psychedelic Research, University of California San Diego, La Jolla, CA 92093, USA. ✉e-mail: jwallach@sju.edu; ahalberstadt@health.ucsd.edu; jmccorvy@mcw.edu

psychedelic from therapeutic properties[5,6]. Nevertheless, fundamental questions exist regarding which receptors and signaling pathways mediate effects of psychedelics, limiting the rational design of new drugs.

Serotonergic psychedelics are derived from multiple chemical scaffolds (e.g., tryptamines, phenethylamines, and lysergamides), all which activate the 5-HT$_{2A}$ receptor (5-HT$_{2A}$R), a G protein-coupled receptor (GPCR). 5-HT$_{2A}$Rs appear to primarily mediate psychedelic experiences supported by evidence that the 5-HT$_{2A}$R antagonist ketanserin attenuates subjective effects of psilocybin and LSD in humans[7,8]. Psychedelics have also been studied in preclinical behavioral models, including the head-twitch response (HTR), which is a 5-HT$_{2A}$R-mediated involuntary head movement in mice that predicts human psychedelic activity[9]. Multiple receptors, however, appear to contribute to the behavioral effects of psychedelics, including psilocybin[10] and LSD[11], which adds to their complicated psychopharmacology.

Ligands targeting GPCRs stabilize certain receptor conformations that energetically favor coupling to transducer proteins[12,13]. Ligand-dependent bias has important implications for drug development and clinical pharmacology[14,15]. For example, the G protein-biased μ-opioid receptor (MOR) agonist oliceridine reportedly has a correspondingly improved tolerability profile[16], although alternative explanations exist[17]. As existing psychedelics activate both Gq and β-arrestin2 via 5-HT$_{2A}$R[18,19], the role these pathways play in the effects of psychedelics is unclear. Although putative non-psychedelic 5-HT$_{2A}$R agonists exist[5,6], an adequate explanation for lack of psychedelic action does not exist. A clear and defined pharmacological signaling mechanism explaining why certain 5-HT$_{2A}$R agonists lack psychedelic effects is thus needed.

Herein, we utilized structure-inspired design to develop 5-HT$_{2A}$R-selective ligands acting as biased agonists and leveraged these to develop a mechanistic and molecular explanation of biased 5-HT$_{2A}$R agonism. Furthermore, these compounds were used to probe the relationship between 5-HT$_{2A}$R-Gq versus 5-HT$_{2A}$R-β-arrestin activity and psychedelic potential in vivo. Our goal was to identify the 5-HT$_{2A}$R transducers mediating psychedelic activity for the rational design of next-generation 5-HT$_{2A}$R agonists.

## Results

### Psychedelics exhibit similar Gq and β-arrestin2 activity at 5-HT$_{2A}$R

To investigate the 5-HT$_{2A}$R signaling profiles of classical psychedelics, we used bioluminescence resonance energy transfer approaches (BRET, Fig. 1A, B, Supplementary Table 1), which provide a proximity measure of intracellular transducer engagement. BRET is used extensively to quantify GPCR-biased agonism and has the advantage of not being susceptible to second messenger amplification or receptor reserve issues obfuscating GPCR signaling preference determinations[20]. BRET has been used to measure 5-HT$_{2A}$R-β-arrestin2 and 5-HT$_{2A}$R-Gq activity directly[11], and to confirm 5-HT$_{2A}$R G protein coupling preferences[18]. In our assay platform, 5-HT$_{2A}$R strongly couples to Gq/11 and β-arrestin2 over all other G protein subtypes and β-arrestin1 (Fig. 1C).

Next, we tested classical psychedelics from multiple chemical classes and examined their effects on Gq dissociation and β-arrestin2 recruitment (Fig. 1D-K). Kinetic issues can confound studies of ligand-directed bias[21], and slow ligand kinetics can delay full receptor occupancy, as observed for LSD at 5-HT$_{2A}$R[19]. Therefore, we thoroughly assessed Gq and β-arrestin2 activities at various time points at 37 °C to ensure full receptor occupancy and confirm that transducer preferences do not substantially change when compared at the same time point. Our results show that psychedelics exhibit dynamic, time-dependent profiles of Gq and β-arrestin activity that in some cases exceed the activity of 5-HT (i.e., superagonism) at longer time points

(e.g., 300 minutes, Supplementary Fig. 1A-H), which is not surprising given the dynamic temporal nature of GPCR signaling[22]. For all tested psychedelics, however, effects on Gq and β-arrestin activity were strikingly similar at equivalent time points and closely mirrored the pathway-balanced endogenous agonist 5-HT (Fig. 1L), indicating no strong preferences for one of the transducers. Psilocin, DMT and 2C-I showed slightly less β-arrestin2 efficacy compared to Gq activity, but the difference was not substantial when both transducers were compared at each respective time point (Supplementary Fig. 1B, C, E). Importantly, all tested psychedelics lacked a strong preference for Gq or β-arrestin2, demonstrating these compounds are not substantially biased for either transducer.

### Rational design of a 5-HT$_{2A}$-selective agonist template

To engineer a series of biased agonists for interrogating the 5-HT$_{2A}$R-coupled signaling pathways associated with psychedelic potential, a scaffold exhibiting some degree of selectivity for 5-HT$_{2A}$R over other 5-HT receptors is required. Most psychedelics are not selective for 5-HT$_{2A}$R, exhibiting complex polypharmacology[11]. 5-HT$_{2A}$R shares considerable homology with 5-HT$_{2B}$R and 5-HT$_{2C}$R, making it challenging to develop selective 5-HT$_{2A}$R agonists. To date, few 5-HT$_{2A}$R-selective agonists have been discovered, but the N-benzyl-phenethylamines 25CN-NBOH[23] and DMBMPP[24] are purported examples. Importantly, recently discovered "non-psychedelic" 5-HT$_{2A}$R agonists show little selectivity for 5-HT$_{2A}$R[5,6], complicating attempts to interpret their psychopharmacology.

To develop selective 5-HT$_{2A}$R biased ligands, we focused on the phenethylamine scaffold, which tends to have high selectivity for 5-HT$_2$ subtypes. 25N or 2C-N (1) was selected as the core phenethylamine based on previous reports of potential 5-HT$_{2A}$R biased agonism[25,26]. We confirmed 25N (1) is a high-affinity, potent 5-HT$_2$ agonist with weak selectivity for 5-HT$_{2A}$R and 5-HT$_{2C}$R over 5-HT$_{2B}$R (Fig. 2A, Supplementary Table 8). Because N-benzylation can increase affinity and potency of phenethylamines at 5-HT$_{2A}$R[27], we converted 25N (1) to 25N-NB (2) (see Supplementary Information), which reduced 5-HT$_{2B}$R efficacy substantially (E$_{MAX}$ = 32% of 5-HT, Fig. 2A). Unfortunately, 25N-NB (2) retained potent 5-HT$_{2C}$R activity (EC$_{50}$ = 8.3 nM), resulting in a weakly (4-fold) 5-HT$_{2A}$R-selective ligand.

To optimize 5-HT$_{2A}$R affinity and selectivity, we synthesized a series of 25N-NB analogs designed to modify the electrostatic properties of the N-benzyl ring system (Supplementary Fig. 2, 12; Supplementary Table 2; Supplementary Data 1, 2, and 3). The increased 5-HT$_{2A}$R affinity of N-benzyl-phenethylamines is thought to result in part from hydrogen bonding between the N-benzyl 2-position and residues in 5-HT$_{2A}$R[28], but we developed an alternate hypothesis that ring-electrostatics (i.e. the effect of increasing π-electron density in portions of the N-benzyl-ring, quantified as Hammett σ constants relative to C$_{5'}$) drives 5-HT$_{2A}$R affinity (Fig. 2B, Supplementary Fig. 3; Supplementary Table 3). Using the 25N series, we found that estimates of increasing electron density around the N-benzyl C$_{5'}$ position (para to the 2'-position) increase 5-HT$_{2A}$R binding affinity and agonist potency (Supplementary Fig. 3; Supplementary Tables 3-8). To confirm the importance of this ring-region, we tested the effect of adding a methoxy group to the C$_{5'}$ position of 25N-NBOMe (4), which this analog 25N-NB-2,5-DiMeO (20) had reduced affinity 400-fold, suggesting steric clash and/or altered electronics disrupted the optimal electrostatic interaction (Supplementary Fig. 3J). The electrostatic-relationship was supported by correlations with additional potency estimates, experimental NMR chemical shifts, and in silico Hirshfeld surface analyses (Supplementary Fig. 3A-I, K-P, Supplementary Tables 4-6). This resulted in the discovery of several high-affinity 5-HT$_{2A}$R agonists within the N-benzyl-phenethylamine class of psychedelics (Supplementary Tables 7, 8, 13).

We then examined the effect of N-benzyl 2-postion substitutions on 5-HT$_{2A}$R selectivity in the 25N series. In the 2-halogen series, a

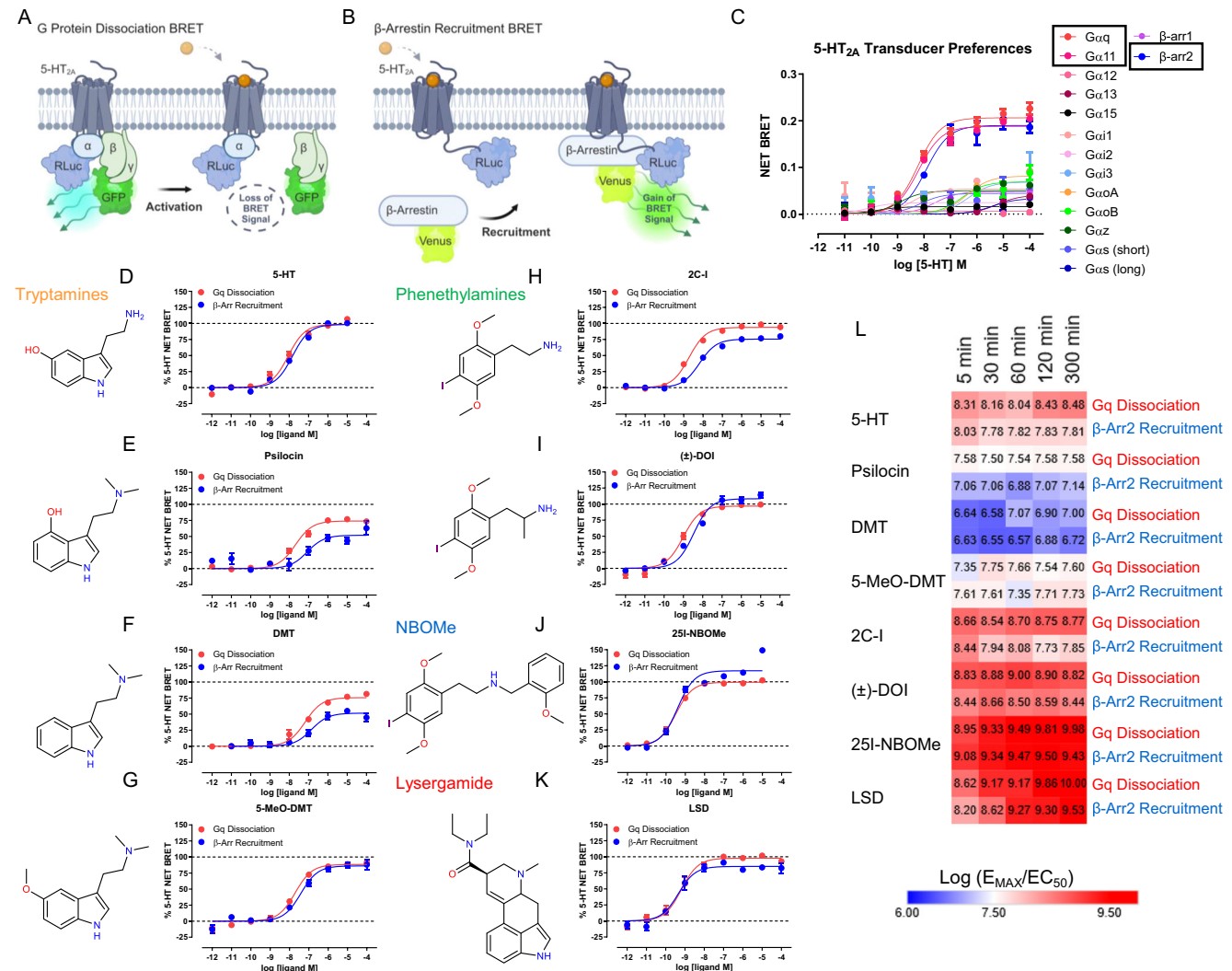

**Fig. 1 | Psychedelics exhibit similar Gq and β-arrestin2 activity at 5-HT$_{2A}$R.** Schematic of 5-HT$_{2A}$ receptor G protein dissociation (**A**) and β-arrestin recruitment (**B**) to determine transducer preferences. (**C**) Determination of 5-HT$_{2A}$ receptor G protein-wide and β-arrestin1/2 transducer preferences as estimated by magnitude net BRET for each transducer as stimulated by 5-HT. Data represent the mean and SEM from three independent experiments, which were performed at 37 °C with 60-minute compound incubations. **D-K** Comparison of 5-HT$_{2A}$ receptor Gq dissociation (red) and β-arrestin2 recruitment (blue) for 5-HT (**D**) and several classes of psychedelics: (**E**) Psilocin, (**F**) DMT, (**G**) 5-MeO-DMT, (**H**) 2C-I, (**I**) DOI, (**J**) 25I-NBOMe, **K** LSD. Data represent the mean and SEM from three independent experiments, which were performed at 37 °C with 60-minute compound incubations. **L** Heat map of Gq dissociation and β-arrestin2 recruitment kinetics displayed as log (E$_{MAX}$/EC$_{50}$) for the psychedelics tested. Data represent the mean and SEM from three independent experiments, which were performed at 37 °C at the indicated compound incubation time points. Source data are provided as a Source Data file.

larger/bulkier 2-bromo or 2-iodo atom on the *N*-benzyl ring reduced 5-HT$_{2C}$R and 5-HT$_{2B}$R activity substantially but retained potent 5-HT$_{2A}$R activity, resulting in increased 5-HT$_{2A}$R selectivity (Fig. 2C, Supplementary Fig. 4A). We compared 25N-NBI (**10**), the most selective 5-HT$_{2A}$R agonist from this series, to the purported 5-HT$_{2A}$R-selective agonist 25CN-NBOH[29] and determined 25N-NBI (**10**) showed 23-fold selectivity for 5-HT$_{2A}$R over 5-HT$_{2C}$R, whereas 25CN-NBOH shows only 7-fold selectivity (Fig. 2D). We confirmed the superior 5-HT$_{2A}$R selectivity of 25N-NBI (**10**) using an orthogonal assay measuring Gq-mediated calcium flux (Supplementary Fig. 4B; Supplementary Table 7), confirming in two functional assays that 25N-NBI (**10**) has greater 5-HT$_{2A}$R selectivity than 25CN-NBOH. 25N-NBI (**10**) also showed selectivity for 5-HT$_{2A}$R over 5-HT$_{2B}$R and 5-HT$_{2C}$R in competitive binding studies (Supplementary Table 8). Furthermore, when tested at >40 additional 5-HT receptors and off-targets, 25N-NBI (**10**) exhibited weak micromolar affinity for all targets screened (Supplementary Table 9). Competitive radioligand binding studies showed high selectivity for 5-HT$_2$ subtypes across the series (Supplementary Tables 8-10). Finally, 25N-NBI (**10**) induced the HTR in mice (ED$_{50}$ = 10.9 μmol/kg), confirming selective 5-HT$_{2A}$R agonists possess psychedelic potential (Supplementary Fig. 4C).

Although a 2-iodo *N*-benzyl substitution yielded a 5-HT$_{2A}$R-selective agonist with psychedelic potential, 25N-NBI (**10**) did not show a preference for either 5-HT$_{2A}$R-Gq or 5-HT$_{2A}$R-β-arrestin2 activity (Supplementary Fig. 4D). Therefore, we focused on another 25N analog, 25N-NB-2-OH-3-Me (**18**; Supplementary Fig. 2), which was rationally designed to maximize C$_{5'}$ electron density (Supplementary Fig. 3A, Supplementary Tables 3, 4) and confirmed to have the highest 5-HT$_{2A}$R affinity in the 25N series (Fig. 2B, E). 25N-NB-2-OH-3-Me (**18**) showed a selective reduction in 5-HT$_{2A}$R Gq-efficacy but not 5-HT$_{2A}$R-β-arrestin2 efficacy compared to 25N-NBOH (**3**) (Fig. 2E), which lacks 3-methyl substitution, providing a critical clue that steric effects in the 3-position influence 5-HT$_{2A}$R biased-agonist activity.

To gain additional insights into the binding mode of the 25N series, we docked 25N-NBI (**10**), 25N-NB-2-OH-3-Me (**18**), and other analogs into the Gq-bound 5-HT$_{2A}$R cryo-EM structure (6WHA)[18] using

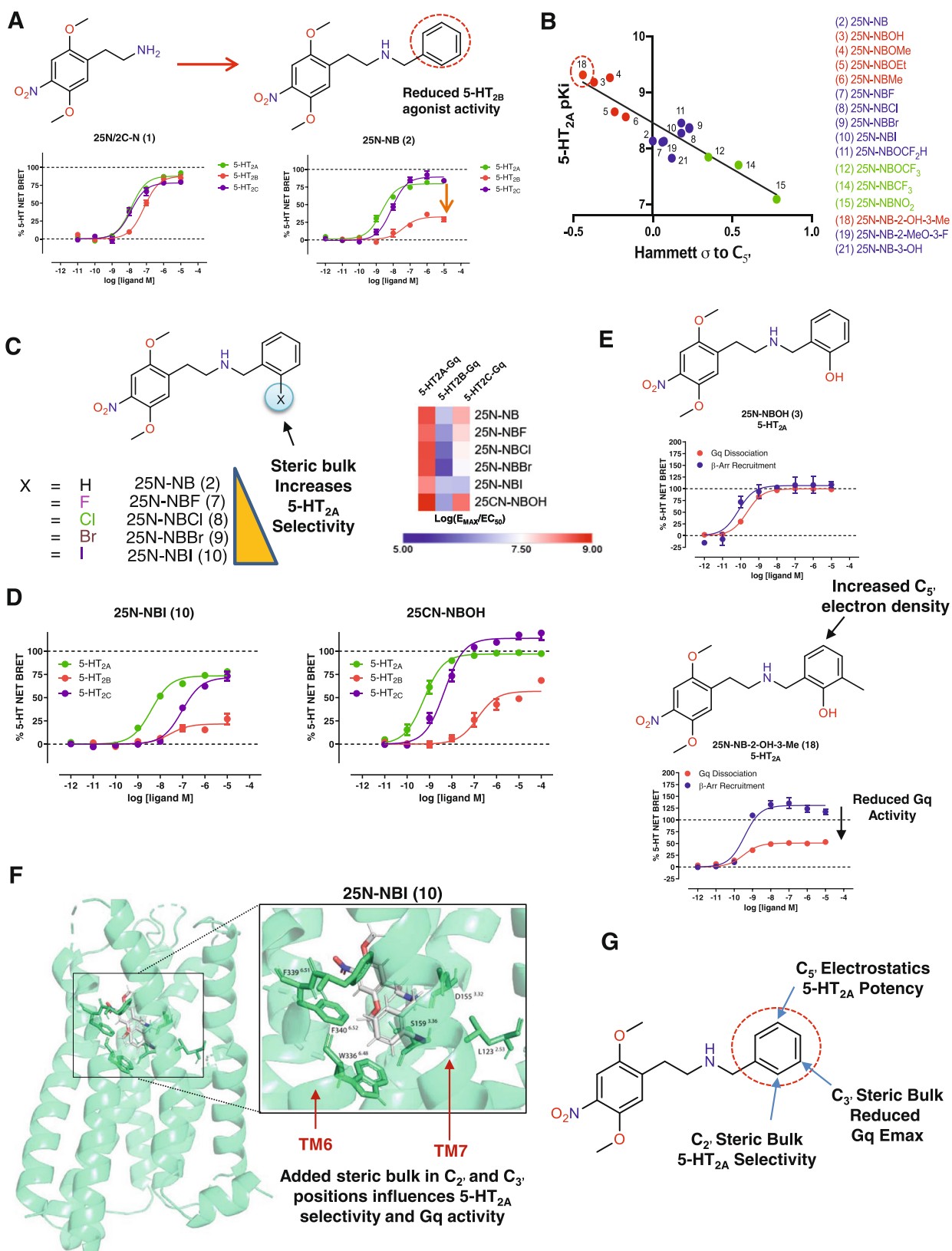

induced fit docking (IFD) (Fig. 2F; Supplementary Fig. 5). Consistent with other studies[18,27], the *N*-benzyl groups formed edge-to-face π-π stacking interactions with F339[6.51] and F340[6.52], known drivers of high 5-HT$_{2A}$R affinity[27,30] (Fig. 2D and Supplementary Table 20). We validated this docking pose and the 5-HT$_{2A}$R model using a series of binding pocket mutants. Examination of mutants in transmembrane 6 (TM6), extracellular loop 2 (EL2), TM3, and TM7 indicate the binding poses are consistent with the 25CN-NBOH-bound cryo-EM structure. Specifically, mutants of key aromatic anchor residues, F339[6.51]L and F340[6.52]L, showed complete loss of 25N-NBI **(10)** Gq activity (Supplementary Fig. 4E), suggesting the *N*-benzyl moiety is positioned near TM6. In summary, we found that 5-HT$_{2A}$R affinity, selectivity, and

**Fig. 2 | Rational design of a 5-HT$_{2A}$-selective agonist template. A** *N*-Benzylation of 25N (**1**) to 25N-NB (**2**) leads to reduced 5-HT$_{2B}$ receptor efficacy, as measured by Gq dissociation by BRET. Data represent the mean and SEM from three independent experiments performed at 37 °C with 60-minute compound incubation. **B** Role of *N*-benzyl ring electrostatics in 5-HT$_{2A}$ receptor potency leading to development of 25N-NB-2-OH-3-Me (**18**) using QSAR correlation between 5-HT$_{2A}$ receptor p$K_i$ and Hammett σ constant values (Pearson's $R = -0.8887$, $R^2 = 0.7897$, 2-tailed $p < 0.0001$, $N = 15$). **C** The relationship between steric bulk and 5-HT$_{2A/2C}$ receptor selectivity is shown for the halogen series, leading to the identification of the 5-HT$_{2A}$ receptor-selective agonist 25N-NBI (**10**) (*left*). Also shown is a 5-HT$_{2A/2C}$ receptor selectivity heatmap comparing the 25N halogen series to 25CN-NBOH (*right*). **D** Comparison of 5-HT$_{2A}$ receptor (green) 5-HT$_{2B}$ receptor (red) and 5-HT$_{2C}$ receptor (purple) Gq dissociation activities for 25N-NBI (**10**) (*left*) and 25CN-NBOH

(*right*). Data represent mean and SEM from three independent experiments, which were performed at 37 °C with 60-minute compound incubation. **E** 5-HT$_{2A}$ receptor Gq dissociation and β-arrestin2 BRET concentration response curves for 25N-NBOH (**3**, *top*) and 25N-NB-2-OH-3-Me (**18**, *bottom*) showing addition of a 3-methyl group leads to reduced Gq-efficacy. Data represent mean and SEM from three independent experiments, which were performed at 37 °C with 60-minute compound incubation. **F** 25N-NBI (**10**) induced fit docking (IFD) with orthosteric site residue side chains displayed. The window shows a zoom-in view illustrating key ligand-residue interactions within the orthosteric site and illustrating the close proximity of the 2'- and 3'-positions to TM6 and TM7, which are known to influence ligand bias. **G** Summary of structure-activity relationships (SAR) for the 25N series encompassing key effects on electrostatics, 5-HT$_{2A}$ receptor selectivity, and reduced Gq E$_{MAX}$. Source data are provided as a Source Data file.

potentially biased agonism can be modulated in the 25N series, and that additional bulk in the 2- and 3-positions of the *N*-benzyl ring may drive 5-HT$_{2A}$R selectivity and ligand bias, respectively (Fig. 2E-G).

## Structure-based design of β-arrestin-biased 5-HT$_{2A}$ agonists

Based on the promising results obtained with the initial 25N series that led to increased 5-HT$_{2A}$R selectivity, we explored whether it is possible to reduce Gq signaling further by replacing the *N*-benzyl ring with bulkier bi-aryl ring systems such as *N*-naphthyl (25N-N1-Nap (**16**)) and *N*-biphenyl (25N-NBPh (**17**)). Steric interactions within GPCR binding pockets are an important driver of agonist, antagonist, and biased-agonist conformational states, especially when they involve residues in the extended binding pocket encompassing TM6 and TM7[12,31]. Strikingly, 25N-N1-Nap (**16**) and 25N-NBPh (**17**) showed a substantial reduction in 5-HT$_{2A}$R Gq-efficacy yet preserved 5-HT$_{2A}$R β-arrestin2-efficacy, thus exhibiting β-arrestin-biased agonism at 5-HT$_{2A}$R compared to the balanced agonist 25N-NBOMe (**4**) (Fig. 3A; Supplementary Table 13). We further explored the SAR by synthesizing *N*-benzyl analogs containing other 2- and 3-substituents and this also results in β-arrestin bias (Supplementary Fig. 6A). Similar to our findings with 25N-NBI (**10**), the bulky analogs 25N-N1-Nap (**16**) and 25N-NBPh (**17**) showed weaker Gq activity at 5-HT$_{2B}$R and 5-HT$_{2C}$R, retaining degrees of 5-HT$_{2A}$R selectivity (Supplementary Fig. 6B; Supplementary Tables 13 and 14). Binding assays confirmed these compounds exhibit weak affinities for other 5-HT receptors and off-targets (Supplementary Table 8-10). To test whether the β-arrestin bias was specific for 5-HT$_{2A}$R, we measured β-arrestin2 activity at 5-HT$_{2B}$R and 5-HT$_{2C}$R and found substantially weaker β-arrestin2 recruitment at these subtypes (Supplementary Fig. 6C), confirming a lack of biased agonism at these receptors. Furthermore, weak activity by β-arrestin biased analogs was confirmed in 5-HT$_{2A}$-G11 dissociation, 5-HT$_{2A/2B/2C}$-Gq/11-mediated calcium flux assays, and at all human 5-HT GPCRs as measured by G protein dissociation (Supplementary Fig. 6D–F). Interestingly, the β-arrestin-biased agonists showed partial antagonism of 5-HT$_{2A}$-Gq activity, similarly found with the non-psychedelic 2-Br-LSD[11], and also showed baseline inhibition in 5-HT$_{2A}$R Gq/11-mediated calcium flux traces at longer time points, suggestive of potential β-arrestin2-mediated desensitization in this cell system (Supplementary Fig. 6E, G). We also confirmed and validated β-arrestin2 bias using our previously established kinetic BRET assay, which consistently showed robust β-arrestin2 and weak Gq activities for 25N-N1-Nap (**16**) and 25N-NBPh (**17**) at all time points tested (Supplementary Fig. 6H).

Encouraged by this validation, we examined the binding mode of 25N-N1-Nap (**16**) at the Gq-bound cryo-EM 5-HT$_{2A}$R structure (6WHA) using IFD (Supplementary Fig. 5). Binding poses using RMSD as a metric were clustered to reveal key interactions with the conserved salt bridge to Asp155[3.32] placing 25N-N1-Nap (**16**) in a similar pose as the 25CN-NBOH structure (Supplementary Table 20; Supplementary Fig. 5). Importantly, in flexible docking, the *N*-naphthyl ring of 25N-N1-Nap (**16**) is wedged near W336[6.48], a highly conserved residue in class A GPCRs (Supplementary Fig. 5). W336[6.48], sometimes called the "toggle switch", is implicated in GPCR activation and biased agonism,

with distinct W6.48 rotamer conformations occurring in activated and non-activated GPCR structures[32–35], including the 25CN-NBOH cryo-EM structure[18]. Given the close proximity and added steric bulk observed in this region with biased agonists like 25N-N1-Nap (**16**), and based on the electrostatic SAR, we hypothesized the weak Gq activity and β-arrestin bias are dependent, in part, on interactions with W336[6.48].

GPCR crystal and cryo-EM structures only show a "snapshot" of activation and are highly dependent on their ternary complex composition and intracellular binding partners. Therefore, we performed molecular dynamics (MD) to investigate interactions between 25N-N1-Nap (**16**) and 5-HT$_{2A}$R, focusing especially on residue W336[6.48], and compared the results to simulations performed with 25CN-NBOH in the cryo-EM structure (Supplementary Data 4-7). Simulations of 250 ns were carried out for each ligand, following a common protocol for system preparation and equilibration. In addition, twelve replicate trajectories were computed for each ligand, initiated from the same starting structures, but with independent assignments of initial random atomic velocities, allowing for expanded sampling of configurations and independent validation of our analyses.

Comparison of the two primary trajectories reveals that the 25N-N1-Nap simulation has features intermediate between the inactive (6WH4) and active (25CN-NBOH) conformations (Fig. 3B, C). For example, in the final frame of the simulation, the outward swing of TM6, with 25N-N1-Nap (**16**), relative to the inactive state (6WH4) is less than that of 25CN-NBOH (7.30° vs. 15.49°) (Fig. 3D). A clear difference in the W336[6.48] χ$_2$ angle frequency was observed between the 25CN-NBOH and 25N-N1-Nap simulations, where 25N-N1-Nap (**16**) showed a preference for W336[6.48] χ$_2$ angle down (centered at −15.6°), and 25CN-NBOH showed a preference for W336[6.48] χ$_2$ angle up (66.2°) in the active rotamer conformation (Fig. 3E), similar to the active state cryo-EM structure (6WHA)[18]. The states observed in our primary simulations were reproduced in the replica simulations, with closely similar side-chain angles. Averaging over all replicates, a W336[6.48] χ$_2$ peak is found at 65.2° for 25CN-NBOH, while for 25N-N1-Nap (**16**), a W336[6.48] χ$_2$ peak is found at −21.2°. Interestingly, in the 25N-N1-Nap simulation, W336[6.48] initially toggles further downward into the pocket and mostly remains in this orientation throughout the simulation, likely to accommodate the larger *N*-naphthyl ring in 25N-N1-Nap (**16**), and is distinct from the W336[6.48] χ$_2$ angles in active (6WHA) and inactive-state 5-HT$_{2A}$R structures (Fig. 3C, E; e.g., PDB structures 6WGT, 6A94, 6A93, 6WH4; compare Supplementary Tables 21 and 22).

To verify the role of W336[6.48] in 5-HT$_{2A}$R biased agonism, we constructed conservative mutations, W336[6.48]Y and W336[6.48]L, designed to increase space and accommodate the larger substituents in 25N β-arrestin-biased ligands. Although the W336[6.48]L mutant exhibited reduced Gq dissociation and impaired β-arrestin recruitment (Supplementary Fig. 6I) preventing further experimental use, the W336[6.48]Y mutant showed robust Gq and β-arrestin recruitment and was used in subsequent experiments (Supplementary Fig. 6I). When 25N-N1-Nap (**16**) and 25N-NBPh (**17**) were tested at the W336[6.48] Y mutant, the

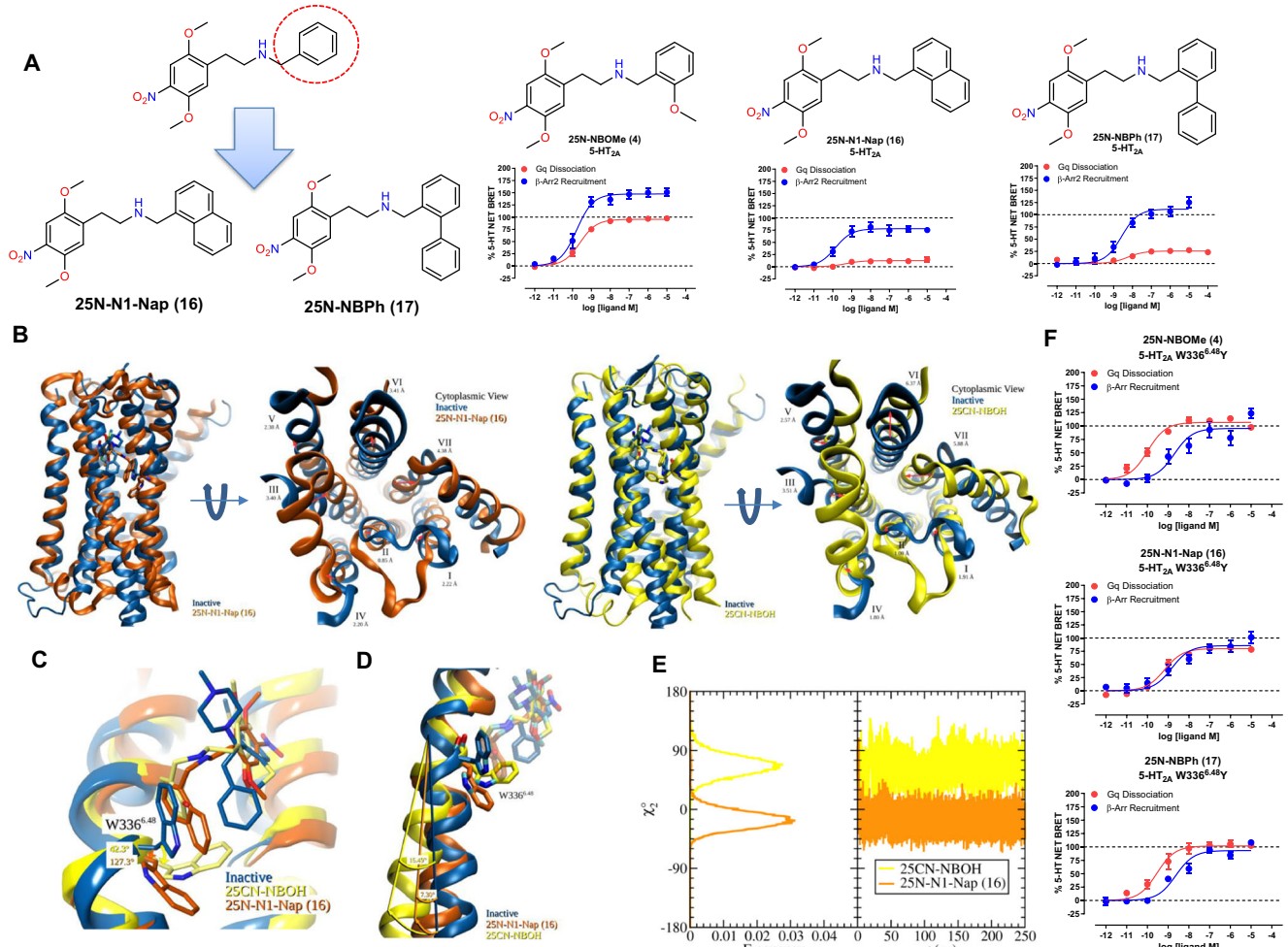

**Fig. 3 | Structure-based design of β-arrestin-biased 5-HT$_{2A}$ agonists. A** Effect of the larger *N*-substituted 25N analogs 25N-N1-Nap (**16**) and 25N-NBPh (**17**) on 5-HT$_{2A}$ receptor Gq dissociation (red) and β-arrestin2 (blue) recruitment. Data represent the mean and SEM from three independent experiments performed at 37 °C with 60-minute compound incubation. **B** Outward pivot of TM6 for 25CN-NBOH and 25N-N1-Nap (**16**) MD simulations relative to the inactive 5-HT$_{2A}$ receptor structure with final frame shown as representative. Note the intermediate TM6 tilt angle with 25N-N1-Nap (**16**) relative to that of 25CN-NBOH. **C** Change in W366$^{6.48}$ toggle switch χ$^2$ angle. χ$^2$ angle given is the absolute difference in peak angle relative to the inactive state. Final frame shown as representative. **D** Distribution and time dependence W366$^{6.48}$ χ$^2$ angle of 25CN-NBOH (yellow) and 25N-N1-Nap (**16**) (orange) simulations. **E** Effect of 5-HT$_{2A}$ receptor W336$^{6.48}$Y mutation on 25N-NBOMe (**4**), 25N-NBPh (**17**), and 25N-N1-Nap (**16**) Gq dissociation (red) versus β-arrestin2 (blue) recruitment activities. Data represent the mean and SEM from three independent experiments performed at 37 °C with 60-minute compound incubation. Source data are provided as a Source Data file.

Gq-efficacy recovered substantially, resulting in balanced agonist activity similar to 25N-NBOMe (**4**) (Fig. 3F). The recovery of Gq activity likely results from the increased space created by a smaller size Tyr residue at position 6.48, allowing the "toggle switch" to move dynamically to initiate Gq-bound activation states. The mutagenesis data support the hypothesis that the W366$^{6.48}$ toggle switch plays an important role in the biased agonism of 25N-N1-Nap (**16**) and 25N-NBPh (**17**), with steric bulk on the *N*-benzyl ring forcing W336$^{6.48}$ into a unique rotamer conformation, making it less likely that Gq-bound conformations will arise while preserving β-arrestin-preferring conformational states.

## β-Arrestin-biased 5-HT$_{2A}$ agonists lack psychedelic potential

The HTR is commonly used as a behavioral proxy for psychedelic effects because non-psychedelic 5-HT$_{2A}$R agonists do not induce head twitches[36], and because there is a robust correlation between HTR activity in mice and potency to induce psychedelic effects in humans and discriminative stimulus effects in rats[9]. To assess psychedelic potential for the 25N series, seventeen compounds were tested in the HTR assay based on their diverse range of Gq and β-arrestin efficacies. Male mice were used to maintain consistency with the validation

experiments supporting use of the HTR as a cross-species readout of psychedelic potential and to leverage the large dataset of HTR dose-response data previously generated[9,36]. Eleven of the compounds increased HTR counts significantly over baseline levels (Supplementary Tables 15 and 16). Not surprisingly, 25N-NBOMe (**4**), which acts as a psychedelic in humans[37], produced a potent HTR response consistent with this effect (Fig. 4A). By contrast, the β-arrestin2-biased compounds 25N-N1-Nap (**16**), 25N-NBPh (**17**) and 25N-NB-2-OH-3-Me (**18**) failed to induce the HTR (Fig. 4B-D; Supplementary Tables 15, 16), suggesting Gq-efficacy is necessary for psychedelic potential. To confirm the β-arrestin2-biased agonists can partition into the brain and engage central 5-HT$_{2A}$R necessary for the HTR, we pretreated mice with those compounds, which subsequently blocked the HTR induced by psychedelic 5-HT$_{2A}$R agonist DOI (Fig. 4E-H). 25N-NBPh (**17**) had lower potency than 25N-N1-Nap (**16**) in the blockade experiments, which we believe is due to pharmacokinetic differences limiting its CNS distribution, potentially reflecting its higher cLogP (4.8 vs. 4.5, respectively), thus we focused on 25N-N1-Nap (**16**) in subsequent experiments. As expected, 25N-N1-Nap (**16**) and 25N-NBPh (**17**) antagonized 5-HT$_{2A}$R-Gq signaling in vitro with potency similar to the

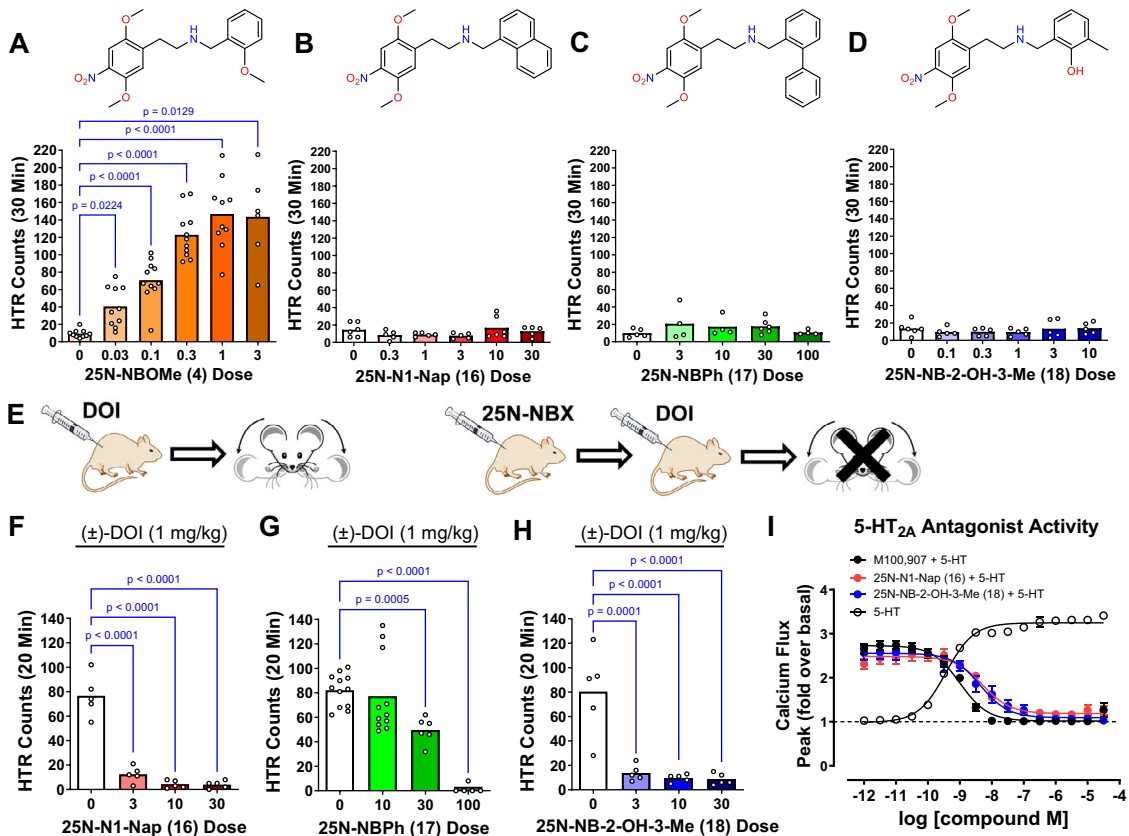

**Fig. 4 | β-Arrestin-biased 5-HT$_{2A}$ agonists lack psychedelic potential. A** 25N-NBOMe (**4**) induces the head-twitch response (HTR) ($W_{5,19.46} = 84.56$, $p < 0.0001$). **B** 25N-N1-Nap (**16**) does not induce the HTR ($F_{5,26} = 1.39$, $p = 0.2618$). **C** 25N-NBPh (**17**) does not induce the HTR ($F_{4,18} = 0.89$, $p = 0.4903$). **D** 25N-NB-2-OH-3-Me (**18**) does not induce the HTR ($F_{2,25} = 0.55$, $p = 0.7377$). **E** Illustration showing the procedures used to confirm that compounds tested in panels F–H are brain penetrant and capable of engaging 5-HT$_{2A}$ receptors in the CNS of mice. Mice were pretreated with vehicle or drug, (±)-DOI (1 mg/kg IP) was injected 10 minutes later, and then HTR activity was assessed for 20 minutes. **F** Pretreatment with 25N-N1-Nap (**16**) blocks the HTR induced by (±)-DOI ($F_{3,16} = 68.43$, $p < 0.0001$). **G** Pretreatment with 25N-NBPh (**17**) blocks the HTR induced by (±)-DOI ($W_{3,14.64} = 144.6$, $p < 0.0001$). **H** Pretreatment with 25N-NB-2-OH-3-Me (**18**) blocks the HTR induced by (±)-DOI ($F_{3,16} = 18.39$, $p < 0.0001$). $P$ values are provided if there were significant differences between groups (Tukey's test or Dunnett's T3 test). HTR counts from individual male C57BL/6 J mice as well as group means are shown. Drug doses are presented as mg/kg. **I** 5-HT$_{2A}$ receptor Gq-mediated calcium flux activity comparing 5-HT (open circles) antagonist activity for 25N-N1-Nap (**16**) (red), 25N-NB-2-OH-3-Me (**18**) to M100,907 (black circles). Data represent the mean and SEM from three independent experiments. Source data are provided as a Source Data file.

5-HT$_{2A}$R-selective antagonist M100907 (Fig. 4I). Given the potential for species differences, which is evident for rodent versus human 5-HT$_{2A}$R[38,39], we verified that β-arrestin bias exhibited by 25N-N1-Nap (**16**) and 25N-NBPh (**17**) is preserved at mouse 5-HT$_{2A}$R (Supplementary Fig. 9A).

To confirm β-arrestin-biased 5-HT$_{2A}$R agonists do not induce the HTR and that this profile is not specific to the 25N series, we synthesized and tested *N*-naphthyl and *N*-biphenyl derivatives of other phenethylamine 5-HT$_{2A}$R agonists (Supplementary Fig. 13, 14). Similar to previous β-arrestin-biased compounds, these compounds exhibited weak Gq-efficacy but produced robust β-arrestin2 recruitment efficacy (Supplementary Fig. 9B; Supplementary Table 13-16). When evaluated in mice, 25O-N1-Nap (**28**) and 2C2-N1-Nap (**29**) failed to induce head twitches (Supplementary Fig. 9C) but fully blocked the HTR induced by DOI (Supplementary Fig. 9D), indicating brain penetration. In summary, five different β-arrestin2-biased 5-HT$_{2A}$R agonists did not induce the HTR when tested at doses that block the response to a balanced agonist, confirming this strategy can consistently generate β-arrestin2-biased 5-HT$_{2A}$R agonists devoid of psychedelic potential.

### 5-HT$_{2A}$-Gq signaling predicts psychedelic potential

Although the 5-HT$_{2A}$R signaling pathways associated with psychedelic potential have been previously investigated, the results were inconclusive[40–42]. To assess further the involvement of 5-HT$_{2A}$R Gq and

β-arrestin2 activity in the HTR, correlation analyses were performed using the 25N series, which contains a mixture of HTR-active and inactive compounds. For the 25N compounds with Gq and β-arrestin2 data ($n = 14$), there was the robust correlation between HTR magnitude (the maximum number of HTR induced by each drug in counts/minute) and 5-HT$_{2A}$R Gq-efficacy (%5-HT E$_{MAX}$; $R_S = 0.8242$, $p = 0.0005$; Fig. 5A). By contrast, HTR magnitude was not correlated with β-arrestin2 recruitment ($R_S = -0.01538$, $p = 0.9638$; Fig. 5B). Notably, the relationship between 5-HT$_{2A}$R Gq-efficacy and HTR magnitude was nonlinear and none of the 25N derivatives with Gq E$_{MAX}$ values <70% induced the HTR, potentially indicating that 5-HT$_{2A}$R efficacy must exceed a strong Gq-efficacy threshold level in order to induce the behavior. Similar to Gq dissociation data, the magnitude of the HTR induced by 25N derivatives is correlated with their 5-HT$_{2A}$R Gq/11-mediated calcium flux response ($R_S = 0.8175$, $p = 0.0002$; Supplementary Fig. 10A) and only the compounds with Gq-mediated calcium flux efficacy exceeding 70% 5-HT E$_{MAX}$ induced head twitches, providing further evidence that 5-HT$_{2A}$R Gq-efficacy is necessary for psychedelic potential.

We also tested whether a similar relationship exists for drug potencies (ED$_{50}$) in HTR experiments. Although the in vivo potencies of psychedelic drugs are known to be correlated with their in vitro potencies at 5-HT$_{2A}$R[43–45], it is not clear whether a similar relationship exists for the HTR. Establishing a potency relationship

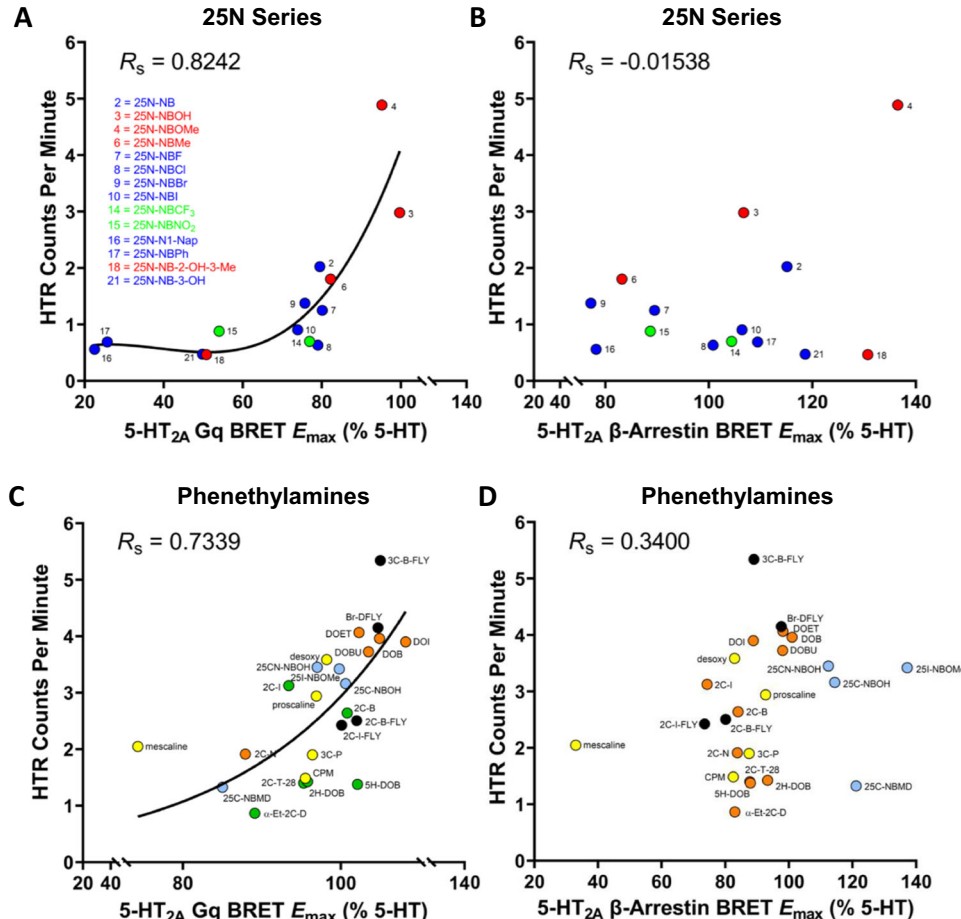

**Fig. 5 | Psychedelic potential is correlated with 5-HT$_{2A}$-Gq signaling. A** Scatter plot showing the relationship between 5-HT$_{2A}$ receptor Gq-efficacy (E$_{MAX}$ values) and head-twitch response (HTR) magnitude (maximum counts per minute induced by the compound) for 14 members of the 25N series. Spearman's rank correlation coefficient $R_s$ is shown. The regression line generated by fitting the data using non-linear regression is included as a visual aid. **B** Scatter plot showing the relationship between 5-HT$_{2A}$ receptor β-arrestin2 recruitment efficacy (E$_{MAX}$ values) and HTR magnitude for 14 members of the 25N series. **C** Scatter plot showing the relationship between 5-HT$_{2A}$ receptor Gq-efficacy (E$_{MAX}$ values) and HTR magnitude for 24 phenethylamine psychedelics. To illustrate the non-linear nature of the relationship, the regression line generated by fitting the data using non-linear regression is shown. **D** Scatter plot showing the relationship between 5-HT$_{2A}$ receptor β-arrestin2 recruitment efficacy (E$_{MAX}$ values) and HTR magnitude for 24 phenethylamine psychedelics. Source data are provided as a Source Data file.

would further support HTR construct validity for predicting human psychedelic action via 5-HT$_{2A}$R. Notably, we observed robust and highly significant correlations between potencies in the HTR assay and in vitro 5-HT$_{2A}$R potency measures, including binding affinities ($K_i$) measured using [$^3$H]-ketanserin, and functional potencies (EC$_{50}$) for activating Gq and β-arrestin2 via 5-HT$_{2A}$R (Supplementary Fig. 10B-F). Similar correlation results were obtained for 5-HT$_{2A}$R Gq and β-arrestin2 pathways, which is not surprising because the rank-order potencies of 5-HT$_{2A}$R agonists for Gq and β-arrestin2 are similar, whereas agonist efficacies often diverge across those two pathways. Overall, these results are consistent with the known role of 5-HT$_{2A}$R in the HTR and show a good correlation between in vitro and in vivo potency measures, but they do not link the HTR to a particular transducer pathway because they fail to account for 5-HT$_{2A}$R efficacy at each respective transducer.

Encouraged by results obtained with the 25N series, we examined whether the magnitude of the HTR produced by other psychedelics is correlated with their 5-HT$_{2A}$R Gq-efficacy. To test this hypothesis, we compared 5-HT$_{2A}$R efficacies and HTR magnitude for 24 phenethylamine psychedelics (Supplementary Fig. 11; Supplementary Tables 11, 12 and 17, 18), which tend to have greater 5-HT$_2$ selectivity compared to other scaffolds[10]. Our results show a significant positive relationship between HTR magnitude and 5-HT$_{2A}$R Gq-efficacy ($R_S$ = 0.7339,

$p < 0.0001$; Fig. 5C). Conversely, there is not a significant correlation between HTR magnitude and 5-HT$_{2A}$R β-arrestin2-efficacy ($R_S$ = 0.34, $p$ = 0.104; Fig. 5D). Consistent with the 25N series data, all of the tested psychedelics activated Gq with relatively high efficacy and the trend-line between HTR magnitude and Gq-efficacy indicates a minimum of 70% 5-HT$_{2A}$R Gq-efficacy (%5-HT E$_{MAX}$). In summary, our results indicate there is a threshold level of 5-HT$_{2A}$R Gq-efficacy required for psychedelic-like activity.

To test the hypothesis that HTR activity and psychedelic potential can be predicted based on the existence of a 5-HT$_{2A}$R Gq-efficacy threshold, we synthesized N-benzyl derivatives of other phenethylamines and made predictions about their HTR activity based on their 5-HT$_{2A}$R Gq-efficacy (Fig. 6A-G). We synthesized and tested 2C2-NBOMe (**31**) (E$_{MAX}$ = 98.4%), 25O-NBOMe (**32**) (E$_{MAX}$ = 99.9%), and 25O-NBcP (**33**) (E$_{MAX}$ = 96.3%), which are highly efficacious 5-HT$_{2A}$R-Gq agonists (Supplementary Fig. 10G-I) and would be predicted to induce the HTR. By contrast, 25D-N1-Nap (**26**) (E$_{MAX}$ = 34.3%), 25O-NB-3-I (**34**) (E$_{MAX}$ = 51.7%), and 25O-NBPh-10'-OH (**35**) (E$_{MAX}$ = 60.8%) have efficacy <70% (Supplementary Fig. 10J-L) and were predicted to be inactive in HTR. Indeed, robust HTR responses were measured for 2C2-NBOMe (**31**), 25O-NBOMe (**32**), and 25O-NBcP (**33**), but not for weaker Gq agonists, 25D-N1-Nap (**26**), 25O-NB-3-I (**34**), and 25O-NBPh-10'-OH (**35**) (Fig. 6B-G; Supplementary Table 19). These results show that the ability

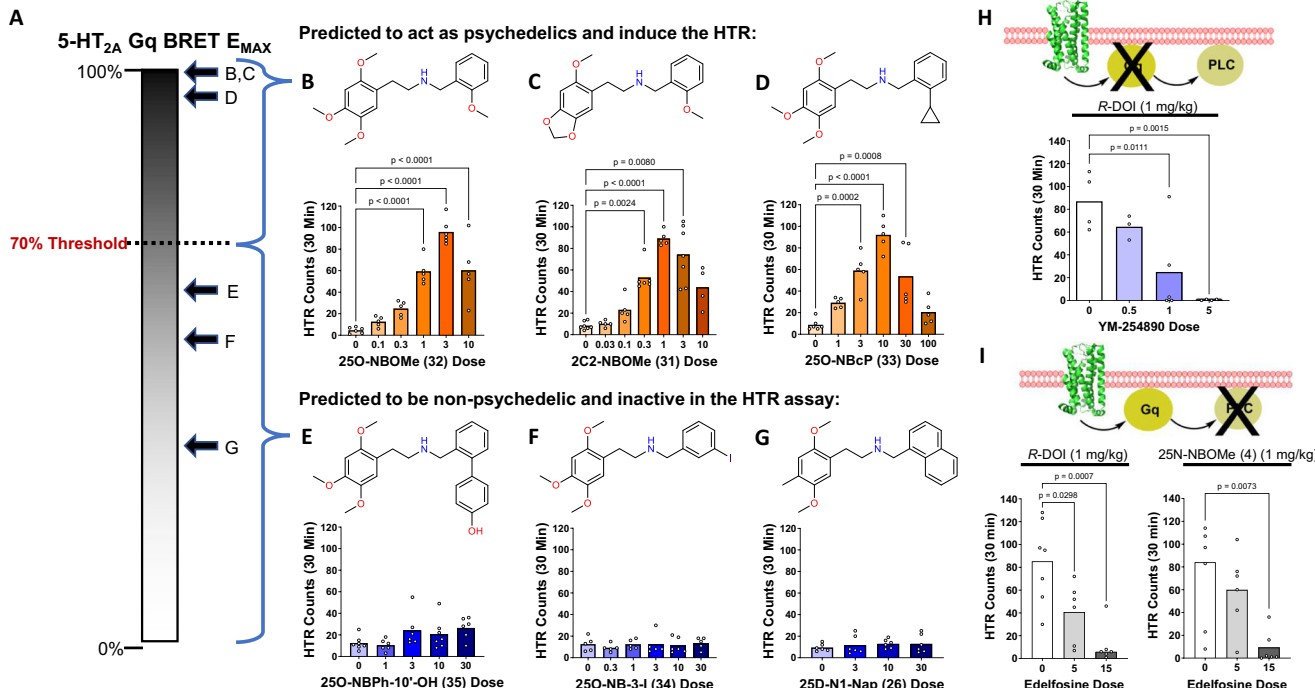

**Fig. 6 | 5-HT$_{2A}$-Gq signaling predicts psychedelic potential. A–G** Activity in the HTR assay can be predicted based on 5-HT$_{2A}$ receptor Gq-efficacy. As predicted, 25O-NBOMe (**32**) ($F_{5,26} = 37.01$, $p < 0.0001$), 2C2-NBOMe (**31**) ($W_{6,12.67} = 96.28$, $p < 0.0001$), and 25O-NBcP (**33**) ($F_{5,25} = 20.57$, $p < 0.0001$) induced head twitches, whereas 25O-NBPh-10'-OH (**35**) ($F_{4,27} = 2.58$, $p = 0.0601$), 25O-NB-3-I (**34**) ($F_{5,25} = 0.26$, $p = 0.9288$), and 25D-N1-Nap (**26**) ($F_{3,20} = 0.37$, $p = 0.7748$) did not induce head twitches. (**H**) Pretreatment with the Gαq/11 inhibitor YM-254890 blocks the HTR induced by R-(-)-DOI ($F_{3,12} = 8.69$, $p = 0.0025$). Mice were treated ICV with YM-254890 or vehicle and then all of the animals received R-(-)-DOI (1 mg/kg IP) 15 minutes later. (**I**) Pretreatment with the phospholipase C (PLC)

inhibitor edelfosine blocks the HTR induced by R-(-)-DOI ($F_{2,16} = 11.34$, $p = 0.0009$) and 25N-NBOMe (**4**) ($F_{2,16} = 6.40$, $p = 0.0091$). Mice received three consecutive injections of vehicle or the indicated dose of edelfosine at 20-minute intervals and then a HTR-inducing drug was administered 10 minutes after the third injection. HTR counts from individual male C57BL/6 J mice as well as group means are shown. *P* values are provided if there were significant differences vs. vehicle control (Tukey's test or Dunnett's T3 test). The dose of YM-254890 is presented in µg; other drug doses are presented as mg/kg. Source data are provided as a Source Data file.

of 5-HT$_{2A}$R agonists to induce head twitches can be predicted based on their 5-HT$_{2A}$R Gq-efficacy.

To directly test the involvement of 5-HT$_{2A}$R-Gq signaling in the HTR, we evaluated whether inhibition of Gq signaling can block the HTR. Because the use of signaling inhibitors can potentially be confounded by off-target effects, we targeted two different proteins within the Gq-PLC effector pathway to generate convergent evidence. YM-254890 selectively inhibits Gq/11[46] and is effective in mice when administered systemically or centrally[47]. Intracranial (ICV) pretreatment with YM-254890 blocked the HTR induced by DOI (Fig. 6H). We also tested the phosphoinositide-selective PLC inhibitor edelfosine[48], which is brain penetrant in mice, with a brain/plasma ratio of ~0.5 after systemic administration[49]. Edelfosine blocked the HTR induced by 25N-NBOMe (**4**) and DOI (Fig. 6I). These results strongly support the conclusion that the HTR induced by psychedelics is dependent on activation of the 5-HT$_{2A}$R-Gq-PLC pathway. This dependence on 5-HT$_{2A}$-Gq signaling is supported by the lack of a HTR with 25N-N1-Nap (**16**) and other β-arrestin2-biased 5-HT$_{2A}$R agonists, and it does not appear that 5-HT$_{2A}$R β-arrestin2 recruitment is sufficient to induce head twitches. Although YM-254890 inhibited the response to DOI to a greater degree than Gq gene deletion[42], the ability of both manipulations to dampen the response to DOI strongly supports our conclusions that 5-HT$_{2A}$R-Gq signaling is necessary for the HTR and therefore psychedelic potential.

### Non-psychedelics do not achieve 5-HT$_{2A}$ Gq-signaling efficacy threshold

The existence of a Gq-efficacy threshold for activity is intriguing because it potentially explains why certain 5-HT$_{2A}$R agonists such as

lisuride fail to induce head twitches in mice and psychedelic effects in humans. To evaluate that possibility, we compared the activity of four psychedelic and four non-psychedelic 5-HT$_{2A}$R agonists in BRET assays and HTR experiments. Clinical studies have confirmed that lisuride[50–53], 2-Br-LSD[54–56], and 6-F-DET[57] lack psychedelic effects in humans; 6-MeO-DMT has not been evaluated clinically but does not substitute in rats trained to discriminate a psychedelic drug[58]. The psychedelic 5-HT$_{2A}$R agonists LSD, psilocin, 5-MeO-DMT, and DET activate 5-HT$_{2A}$R-Gq signaling with E$_{MAX}$ ranging from 74.1–98.8% (Fig. 7A) and induce the HTR (Fig. 7B). Conversely, the non-psychedelic 5-HT$_{2A}$R agonists lisuride, 2-Br-LSD, 6-F-DET, and 6-MeO-DMT activate 5-HT$_{2A}$R-Gq signaling with E$_{MAX}$ < 70% (Fig. 7A) and do not induce head twitches (Fig. 7B). For the eight compounds, there was a significant positive correlation ($R = 0.8948$, $p = 0.0027$) between HTR magnitude and 5-HT$_{2A}$R-Gq E$_{MAX}$ (Fig. 7C). These results are consistent with the predicted activity threshold and demonstrate that the relationship between HTR magnitude and 5-HT$_{2A}$R Gq-efficacy extends to tryptamines and lysergamides. In summary, lisuride, 2-Br-LSD, 6-F-DET, and 6-MeO-DMT are likely non-psychedelic because of weak 5-HT$_{2A}$R Gq-efficacy.

### In vitro and in vivo effects of β-arrestin-biased 5-HT$_{2A}$ agonists

An intriguing question raised by our results is how β-arrestin-biased 5-HT$_{2A}$R agonists differ from 5-HT$_{2A}$R antagonists. One hallmark of β-arrestin-biased agonists is the induction of arrestin-dependent internalization and downregulation, which contributes to drug tolerance[59]. To determine whether 25N-N1-Nap (**16**) and 25N-NBPh (**17**) induce β-arrestin2-dependent receptor internalization, which would further support their β-arrestin-biased agonist profile, we conducted NanoBit

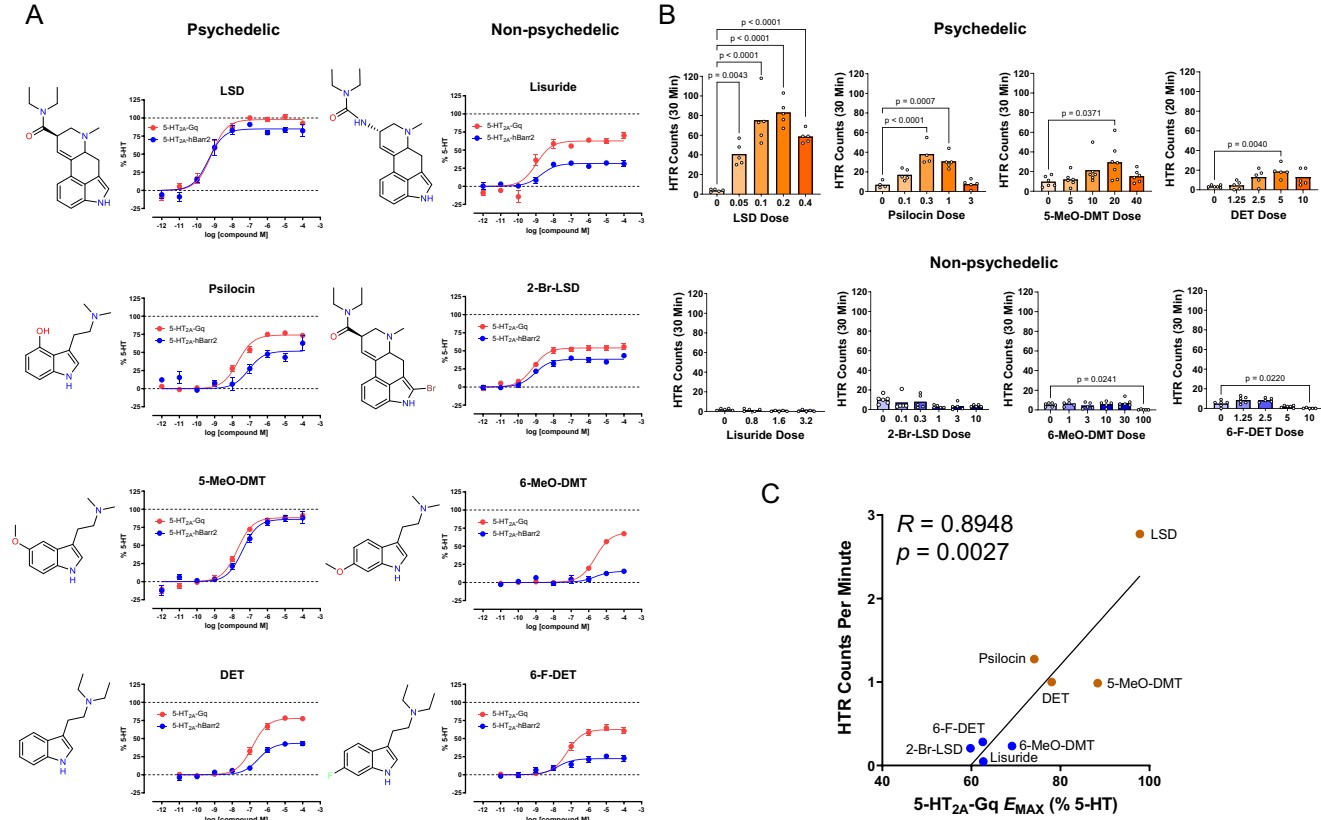

**Fig. 7 | Non-psychedelics do not achieve 5-HT$_{2A}$ Gq-signaling efficacy threshold. A** Effect of (+)-lysergic acid diethylamide (LSD), psilocin, 5-methoxy-*N,N*-dimethyltryptamine (5-MeO-DMT), *N,N*-diethyltryptamine (DET), lisuride, ( + )−2-bromolysergic acid diethylamide (2-Br-LSD), 6-methoxy-*N,N*-dimethyltryptamine (6-MeO-DMT), and 6-fluoro-*N,N*-diethyltryptamine (6-F-DET) on 5-HT$_{2A}$ receptor Gq dissociation (red) and β-arrestin2 (blue) recruitment. The BRET data for LSD, psilocin, and 5-MeO-DMT are from Fig. 1; the data for 2-Br-LSD were published previously[11]. Data represent the mean and SEM from three independent experiments. **B** Effect of LSD, psilocin, 5-MeO-DMT ($F_{4,26}$ = 3.09, $p$ = 0.0331), DET, lisuride, 2-Br-LSD, 6-MeO-DMT ($F_{5,26}$ = 4.19, $p$ = 0.0063), and 6-F-DET ($F_{4,23}$ = 13.49,

$p$ < 0.0001) on the head-twitch response (HTR) in mice. The HTR data for LSD, DET, psilocin, lisuride, and 2-Br-LSD were re-analyzed from published experiments[9,11,78,85]. HTR counts from individual male C57BL/6 J mice as well as group means are shown. *P*-values are provided if there were significant differences between groups (Tukey's test). **C** Scatter plot and linear regression showing the correlation between 5-HT$_{2A}$ receptor Gq-efficacy ($E_{MAX}$ values) and HTR magnitude (maximum counts per minute induced by each drug). Pearson's correlation coefficient *R* is shown. The *p*-value is 2-tailed. Drug doses are presented as mg/kg. Source data are provided as a Source Data file.

internalization assays[60,61] that utilize a minimally tagged N-terminal HiBiT-tagged 5-HT$_{2A}$R and a membrane impermeable LgBit Nanoluc complementation fragment to measure surface expression loss (Fig. 8A). In experiments where untagged β-arrestin2 was co-expressed (similar conditions to β-arrestin2 BRET assays), 5-HT and DOI induced a strong internalization after 60 minutes. 25N-N1-Nap **(16)** and 25N-NBPh **(17)** also induced robust internalization, similar to their potencies in β-arrestin2 recruitment assays (Fig. 8B). By contrast, the 5-HT$_{2A}$R antagonist/inverse agonist pimavanserin (PIM) did not induce internalization, suggesting that β-arrestin-biased 5-HT$_{2A}$R agonists are distinct from pimavanserin in their ability to downregulate 5-HT$_{2A}$R. Previous work in whole cell and in vivo systems revealed that some 5-HT$_{2A}$R antagonists cause atypical 5-HT$_{2A}$R internalization[62,63], but other 5-HT$_{2A}$R antagonists such as eplivanserin (SR-46349B) upregulate 5-HT$_{2A}$R in mice[64]. Thus, how these 5-HT$_{2A}$R ligands affect in vivo receptor expression may be influenced by several factors and requires further study.

5-HT$_{2A}$R agonists are known to cause receptor downregulation and tolerance or tachyphylaxis[65]. We tested whether β-arrestin2-biased 5-HT$_{2A}$R agonists induce tolerance in vivo, based on the rationale that tolerance could potentially act as an in vivo readout of β-arrestin2 recruitment. Our hypothesis was that repeated treatment with 25N-N1-Nap **(16)** would induce tolerance similar to the balanced 5-HT$_{2A}$R

agonist DOI. Results were normalized to the vehicle control group to facilitate comparison of the amount of tachyphylaxis induced by the three 5-HT$_{2A}$R ligands. Once daily administration of 25N-N1-Nap **(16)** or DOI to mice for five consecutive days reduced the ability of a challenge dose of DOI administered 24 h later to induce the HTR (Fig. 8C, D). Conversely, the response to DOI was not altered in mice treated repeatedly with a dose of the 5-HT$_{2A}$R antagonist PIM that is capable of blocking the acute behavioral response to DOI (Fig. 8E). Hence, repeated administration of 25N-N1-Nap **(16)**, DOI, and PIM produced effects in mice that closely parallel the in vitro internalization data. Based on these results, β-arrestin2 may play a role in the tachyphylaxis that occurs after repeated administration of 5-HT$_{2A}$R agonists.

Finally, to assess whether β-arrestin-biased 5-HT$_{2A}$R agonists may have similar therapeutic potential as 5-HT$_{2A}$R antagonists/inverse agonists, we tested whether 25N-N1-Nap **(16)** can antagonize PCP-induced hyperactivity in mice. 5-HT$_{2A}$R antagonists are highly effective at blocking the hyperlocomotion induced by NMDA receptor antagonists such as PCP and MK-801[17,66]. 25N-N1-Nap **(16)** blocked the hyperactivity induced by PCP in mice when tested at a dose that had no effect on baseline activity (Fig. 8F). We confirmed that the 5-HT$_{2A}$-selective antagonist M100907 has a similar effect on reducing PCP-induced hyperactivity in mice (Fig. 8G) consistent with an antipsychotic-like profile.

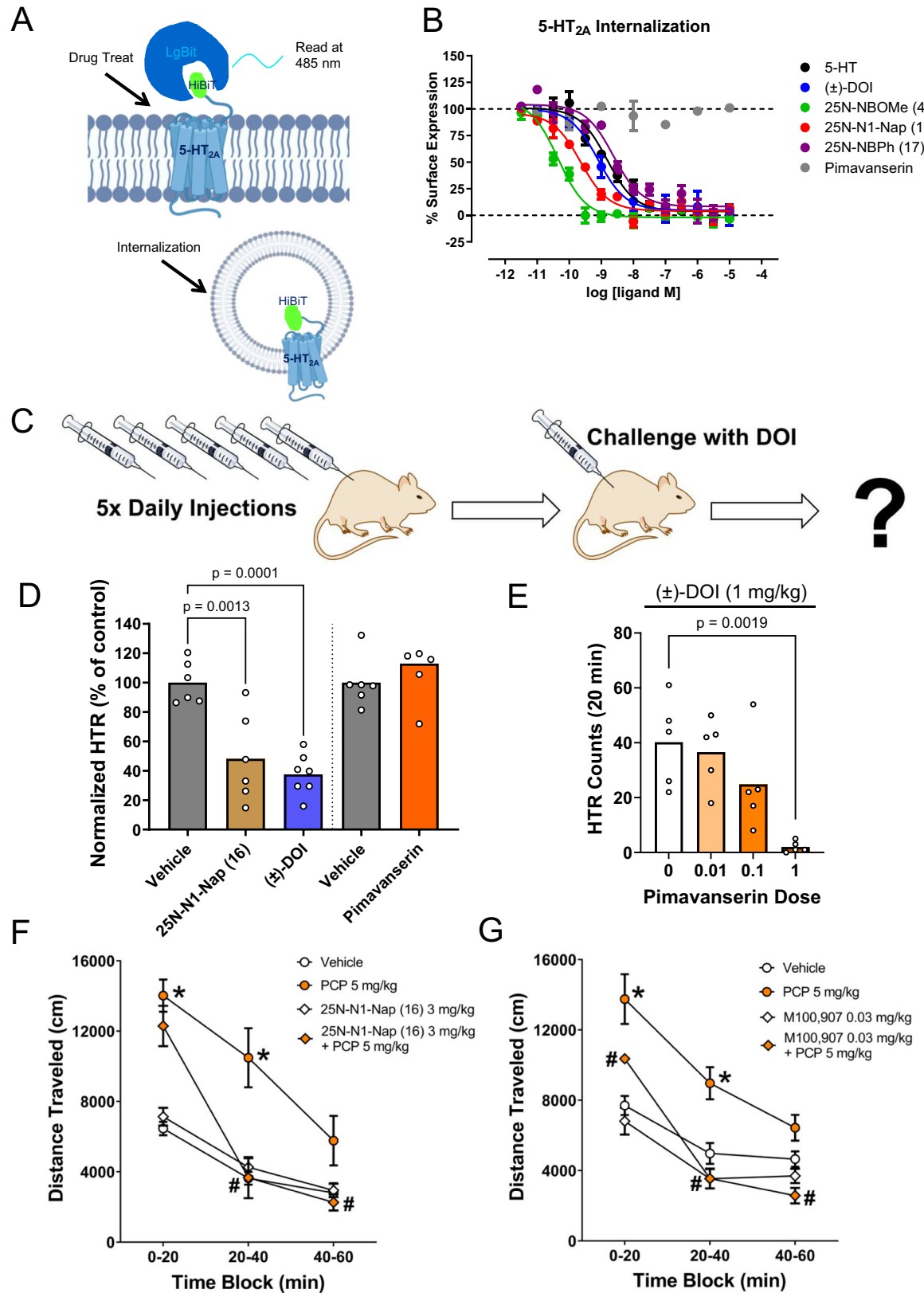

## Discussion

Our ultimate aim was to elucidate which 5-HT$_{2A}$R transducer is responsible for psychedelic activity. Here we show that rational and structure-based design can be leveraged to develop 5-HT$_{2A}$R-selective compounds with a wide-range of functional activities, including β-arrestin2-biased agonists. Docking, MD, and mutagenesis revealed that

W336$^{6.48}$ plays a key role in the mechanism of β-arrestin2-biased agonists. Finally, β-arrestin2-biased 5-HT$_{2A}$R agonists were used to probe the involvement of this transducer toward the psychedelic potential of 5-HT$_{2A}$R agonists. The ability of 5-HT$_{2A}$R agonists to induce the HTR was found to be correlated with Gq-efficacy but not β-arrestin2 recruitment. 5-HT$_{2A}$R β-arrestin2-biased agonists did not induce

**Fig. 8 | In vitro and in vivo effects of β-arrestin-biased 5-HT$_{2A}$ agonists.**
**A** Scheme of NanoBit Internalization Assay for 5-HT$_{2A}$R measuring loss of surface expression with the membrane impermeable LgBit. **B** 5-HT$_{2A}$R internalization concentration response calculated as percent basal surface expression and normalized to 0% of max 5-HT response. Data represent mean and SEM from 3 independent experiments with 5-HT (black), DOI (blue), pimavanserin (grey), 25N-NBOMe **(4)** (green), 25N-N1-Nap **(16)** (red) and 25N-NBPh **(17)** (purple). **C** Cartoon showing the procedures used to induce tolerance in the head-twitch response (HTR) assay. Mice were injected with vehicle or drug once daily for 5 consecutive days and then challenged with (±)-DOI (1 mg/kg IP) 24 hours after the last injection. **D** Repeated administration of (±)-DOI and 25N-N1-Nap **(16)** ($F_{2,16} = 16.68$, $p = 0.0001$), but not pimavanserin ($t_{10} = 1.067$, 2-tailed $p = 0.3111$), induces a tolerance to the HTR induced by DOI. Mice was treated with vehicle, 25N-N1-Nap **(16)** (20 mg/kg/day SC), or (±)-DOI (10 mg/kg/day SC). Additional mice were treated with vehicle or pimavanserin (1 mg/kg/day SC). HTR counts are expressed as a percentage of the response in the respective control group; data from individual male C57BL/6 J mice as well as group means are shown. *P*-values are provided if there were significant differences between groups (Tukey's test). **E** Pimavanserin blocks the response to (±)-DOI injected 20-minutes later ($F_{3,16} = 8.19$, $p = 0.0016$). HTR counts from individual mice as well as group means are shown. *P* values are provided if there were significant differences between groups (Tukey's test). **F** 25N-N1-Nap **(16)** attenuates phencyclidine (PCP)-induced hyperactivity (pretreatment × PCP: $F_{1,20} = 7.09$, $p = 0.015$; pretreatment × PCP × time: $F_{2,40} = 11.03$, $p = 0.0002$). Locomotor activity was measured as distance traveled (cm), presented as group means ± SEM. $N = 6$ male C57BL/6 J mice/group. *$p < 0.05$, significant difference vs. vehicle; #$p < 0.05$, significant difference vs. PCP alone (Tukey's test). **G** M100907 attenuates PCP-induced hyperactivity (pretreatment × PCP: $F_{1,20} = 7.41$, $p = 0.0131$). $N = 6$ male C57BL/6 J mice/group. *$p < 0.05$, significant difference vs. vehicle; #$p < 0.05$, significant difference vs. PCP alone (Tukey's test). Drug doses are presented as mg/kg. Source data are provided as a Source Data file.

psychedelic-like behavioral effects, but blocked psychedelic-like behaviors in vivo. Overall, we identified multiple structural and chemical features of psychedelics that can be targeted to fine-tune 5-HT$_{2A}$R activity, potentially modulating therapeutic efficacy and tolerability of 5-HT$_{2A}$R ligands for various therapeutic indications including psychosis.

Although the specific 5-HT$_{2A}$ transducers mediating the HTR have not been conclusively identified, Gq/11 and β-arrestin2 are clear candidates. Previous studies in constitutive knockout (KO) mice have attempted to address the role of Gq and β-arrestin2 in the HTR but did not yield conclusive results. Gq deletion attenuated but did not eliminate the HTR induced by DOI[42], potentially because other Gq protein subtypes such as G11 may be involved. HTR studies in β-arrestin2 KO mice have variously reported that the response to DOI is unaltered[41,67], response to 5-MeO-DMT is enhanced[68], or response to LSD is attenuated[40]. Although the reason for these discrepancies is unknown, it could reflect altered membrane receptor trafficking in β-arrestin2 KO mice, leading to constitutive receptor desensitization and other adaptations. Given the lack of an effect of 25N-N1-Nap **(16)** and other β-arrestin2-biased 5-HT$_{2A}$R agonists in the HTR paradigm, 5-HT$_{2A}$R β-arrestin2 recruitment does not appear to be sufficient to induce head twitches. Although β-arrestin2 may not be involved in the HTR, our findings do implicate β-arrestin2 in the tachyphylaxis that occurs after repeated administration of 5-HT$_{2A}$R agonists. Mice treated repeatedly with psychedelic drugs develop tolerance and cross-tolerance, potentially reflecting 5-HT$_{2A}$R downregulation[67]. Similarly, repeated treatment with 25N-N1-Nap (16) induced a tolerance to a subsequent challenge dose of DOI, indicating β-arrestin2-biased agonists may also be capable of inducing 5-HT$_{2A}$R downregulation. Although it was recently reported that β-arrestin2 KO mice do show tolerance to DOI-induced HTR after repeated treatment[67], the use of constitutive KOs to address the mechanisms of GPCR tolerance is not ideal because of potential aforementioned adaptations (e.g. β-arrestin1 to offset loss of β-arrestin2).

One important result of these studies is the discovery of an apparent 5-HT$_{2A}$R Gq-efficacy threshold for induction of the HTR in mice. In addition to finding a robust and highly significant correlation between the magnitude of the HTR and multiple readouts of 5-HT$_{2A}$R Gq-efficacy, we found that compounds with a Gq E$_{MAX}$ < 70% do not induce the HTR. 5-HT$_{2A}$R Gq-efficacy was found to be correlated with HTR magnitude in previous studies of mixed 5-HT$_{2A/2C}$ agonists[69,70], but those studies did not test enough compounds to identify a clear Gq-efficacy threshold for HTR activity. Since mice exhibit a baseline level of spontaneous HTR that is driven by basal 5-HT$_{2A}$R activation[71], the existence of a threshold for HTR activity is not surprising because compounds with low Gq-efficacy may act as partial antagonists relative to endogenous 5-HT basal stimulation. A similar threshold may exist for psychedelic effects in humans because all psychedelics we tested activated 5-HT$_{2A}$R-Gq dissociation with E$_{MAX}$ > 70%, whereas non-

psychedelic analogs have lower efficacy (E$_{MAX}$ < 70%), which likely explains why the non-psychedelics do not induce the HTR in mice or psychedelic effects in humans. Consistent with our findings, lisuride (E$_{MAX}$ = 48.6%) was reported to have substantially lower Gq-efficacy than LSD (Emax = 84.6%) or DOI (E$_{MAX}$ = 81.3%) in a 5-HT$_{2A}$R-Gq calcium mobilization assay[72]. Although we did not test the putative non-psychedelic 5-HT$_{2A}$R agonist tabernanthalog, it reportedly has E$_{MAX}$ = 57% in a 5-HT$_{2A}$R calcium flux assay[5], which may be too low to induce the HTR. These data have implications for drug development as it should be possible to identify 5-HT$_{2A}$R-Gq partial agonists that do not induce the HTR and lack strong psychedelic effects in humans but retain sufficient Gq-efficacy to induce therapeutic neurophysiological effects via 5-HT$_{2A}$R (e.g., induction of neuroplasticity). 2-Br-LSD is consistent with this hypothesis because it induces neuroplasticity via 5-HT$_{2A}$R even though its 5-HT$_{2A}$R-Gq E$_{MAX}$ is subthreshold to induce the HTR[11]. Based on these results, it should be possible to rationally design non-psychedelic 5-HT$_{2A}$R agonists with therapeutic potential by fine-tuning 5-HT$_{2A}$R Gq-efficacy. In effect, partial 5-HT$_{2A}$R-Gq agonists may act as mixed agonist-antagonists, similar to buprenorphine and other opioid receptor partial agonists[73], which contribute to their superior safety profile and greater tolerability[17]. Future studies will also need to examine partial 5-HT$_{2A}$R-Gq agonists in preclinical disease models to assess whether they possess therapeutic-like activity, for example, as rapid-acting antidepressants.

Based on our results, ligand interactions with residue W336$^{6.48}$ are an important predictor of agonists with weaker Gq-efficacy but do not fully explain the β-arrestin bias exhibited by these compounds. Residue W6.48 is known to be critical in propagating conformational changes by altering the position of TM6 to impact the cytosolic transducer binding pockets and influence bias[34], but the change in W336$^{6.48}$ in our 25N-N1-Nap **(16)** simulation can likely influence other trigger motifs (PIF and NPxxY), which may be important to preserve GPCR-arrestin recruitment. For example, an influence of W6.48 on the PIF/PIW motif and corresponding changes in TM6 orientation were observed in MD studies with bias at β$_2$-adrenergic[32], rhodopsin[74], MOR[34], 5-HT$_{2C}$R[75], S1PR1[76], and other GPCRs[35]. Moreover, our MD results with the "partially active" TM6 orientation are intriguing given several β-arrestin1 GPCR structures, including LSD-bound 5-HT$_{2B}$R structures[77], show a larger TM6 outward movement relative to the G protein-bound state, though differences may exist in the initial conformation recognized by the transducer and the resulting complex. Therefore, further structural studies are needed to identify other conformational changes involved in Gq versus arrestin-modulation of the binding pocket, especially with respect to these discovered biased ligands.

Several limitations are noted in this study. First, is the use of high-expressing recombinant proteins in immortalized cells, which are necessary for adequate signal detection in vitro. Future studies investigating 5-HT$_{2A}$R transducer effects in native tissues are

warranted. Second, the use of non-human model organisms raises questions about human translatability. We used mouse HTR as a predictor of human psychedelic-like activity because it has excellent predictive and construct validity for psychedelics and can distinguish non-psychedelic 5-HT$_{2A}$ agonists[9,11,78]. However, species nuances may limit extrapolation of these findings to humans and the degree Gq-efficacy thresholds align in humans and rodents is unclear. Finally, we chose not to use the forced swim test or tail suspension to assess antidepressant-like effects with 5-HT$_{2A}$R β-arrestin-biased agonists, because animal behavioral despair models lack translational validity[79–81]. Utilizing an RDoC-like approach[82] to investigate potential clinical applications will best serve to illuminate the therapeutic potential of biased or partial 5-HT$_{2A}$R agonists. The potential therapeutic utility of β-arrestin-biased 5-HT$_{2A}$R agonists needs further study, but our findings suggest they mimic psychedelics in some ways (induction of cross-tolerance and 5-HT$_{2A}$R downregulation) but also produce antagonist-like effects (e.g. blockade of PCP hyperlocomotion and DOI-induced HTR). Potentially, β-arrestin-biased ligands may mimic the therapeutic effects of 5-HT$_{2A}$R antagonists with less potential to disrupt cognition, possibly improving their efficacy and tolerability, but pharmacokinetic parameters should be investigated to ensure proper dosing and receptor occupancy in future experiments.

In conclusion, we have shown that psychedelic drugs activate both 5-HT$_{2A}$R-Gq dissociation and β-arrestin2 recruitment, which led to us to design 5-HT$_{2A}$R-selective biased agonists to probe 5-HT$_{2A}$R transducers necessary for psychedelic potential. These results indicate that a 5-HT$_{2A}$R Gq-efficacy is necessary to produce psychedelic-like effects, as measured by the HTR, and that it is possible to predict psychedelic potential based on this threshold. This study has implications for understanding the neurobiological basis of psychedelic effects and reveals strategies for designing non-psychedelic 5-HT$_{2A}$R agonists that can potentially be used as therapeutics.

## Methods

### Ethical statement
The mouse studies were conducted in accordance with National Institutes Health (NIH) guidelines and were approved by the University of California San Diego Institutional Animal Care and Use Committee (Protocol #S17044).

### Synthesis materials and methods
N-benzyl-compounds were synthesized by reductive amination (using NaBH$_4$) of the pre-formed imine (in solution) obtained by treating the primary amine (e.g., 25N) with the respective aldehyde in dry (3 Å molecular sieves) methanol and THF in the presence of 3 Å molecular sieves in the dark under an argon atmosphere for at least 48 hours. Descriptions, including yields and analytical data including $^1$H and $^{13}$C NMR chemical shift assignments, $^1$H and $^{13}$C NMR spectra, HPLC traces are provided in the Supplementary Information.

Intermediates and reagents for synthesis were purchased from Sigma-Aldrich (St Louis, MO, USA), AKSci, and Alfa Aesar. Reagents were generally 95% pure or greater. 200-proof ethyl alcohol was obtained from Pharmaco (Greenfield Global, CT, USA). Silica gel flash column chromatography was performed using Merck silica gel grade 9385 (230-400 mesh, 60 Å). Melting points were measured using a Digimelt A160 SRS digital melting point apparatus (Stanford Research Systems, Sunnyvale, CA, USA) using a ramp rate of 2 °C/min.

### High-performance liquid chromatography (HPLC)
HPLC analyses were performed on an Agilent 1260 Infinity system that includes a 1260 quaternary pump VL, a 1260 ALS autosampler, a 1260 Thermostatted Column Compartment, and a DAD Multiple Wavelength Detector (Agilent Technologies, Santa Clara, CA, USA). Detection wavelengths were set at 220, 230, 254, and 280 nm but only 220 nm was used for analysis. A Zorbax Eclipse XDB-C18

analytical column (5 μm, 4.6 × 150 mm) from Agilent Technologies was used. Mobile phase A consisted of 10 mM aqueous ammonium formate buffer titrated to pH 4.5. Mobile phase B consisted of acetonitrile. The injection volume of samples was 10 μL, flow rate was 1.0 mL/min, and the column temperature was set at 25ºC. Samples were prepared by preparing a 1 mg/mL solution in 1:1 A:B. All samples were injected in duplicate with a wash in between each run. Run time was 10 minutes with a mobile phase ratio (isocratic) of 1:1 for A:B. Chromatograms were analyzed using the Agilent ChemStation Software (Agilent Technologies). Purity values were calculated from area under the curves of the absorbance at 220 nm of any resulting peaks.

### High resolution mass spectrometry (HRMS)
HRMS data were obtained on a Thermo Orbitrap Exactive Mass Spectrometer with an Orbitrap mass analyzer. The instrument was calibrated using electrospray ionization with Pierce$^{TM}$ LTQ ESI Positive Ion Calibration Solution from ThermoFisher Scientific. Samples were introduced into the instrument and ionized via an Atmospheric Solids Analysis Probe (ASAP). Data were analyzed in the Thermo Xcalibur Qual Browser software and identity was confirmed if <5 ppm error. Measurement parameters were as follows: Aux gas flow rate-8, Spray Voltage-3.50 kV, Capillary temperature-275ºC, Capillary Voltage-25.00 V, Tube Lens Voltage-65.00 V, Skimmer Voltage-14.00 V, Heater Temperature-100ºC.

### Elemental analysis
Elemental analysis (C, H, N) was run on select compounds by Galbraith Laboratories, Inc. (Knoxville, TN).

### Nuclear magnetic resonance
$^1$H (400 MHz) and $^{13}$C NMR spectra (101 MHz) were obtained on the hydrochloride salts as a solution in anhydrous d$_6$-DMSO (~20 mg/mL) (>99.9% D, Sigma-Aldrich) on a Bruker Avance III with PA BBO 400S1 BBF-H-D-05 Z plus probe (Bruker Corporation, Billerica, MA, USA). Solvent (δ = 2.50 and 39.52 ppm for $^1$H and $^{13}$C spectras respectively) was used for internal chemical shift references. $^{19}$F (376.5 MHz) NMR was run as described above using ~100 μL of trichlorofluoromethane (99%+, Sigma-Aldrich) as internal reference (δ = 0.0 ppm). Full NMR chemical shift assignments were made using chemical shift position, splitting patterns, $^{13}$C and $^{13}$C PENDANT or APT and hetero- and homo- 2-D experimentss including HMQC or HSQC, HMBC and COSY (45° pulse tilt). A background water concentration of solvent (determined as integration ratio relative to solvent shift) was also determined to check for water content (indicative of a hydrate) within the final salts for 25N compounds. Evidence of hydrates was not observed.

### Receptor binding experiments
Competition-based receptor binding studies were performed by the National Institute of Mental Health Psychoactive Drug Screening Program (NIMH PDSP) using described methods[83]. Target compounds were dissolved in DMSO and an initial screen performed to assess displacement of the radioligand at target receptors using a concentration of 10 μM. Those compounds that showed >50% displacement of specific radioligand binding for a given receptor then underwent secondary screenings at varying concentrations to determine pK$_i$ values. For the K$_i$ experiments, compounds were run in triplicate on separate plates. Each plate contained a known ligand of the receptor as a positive control. Full experimental details are available in the NIMH PDSP assay protocol book. For all 5-HT subtypes, an $N = 3$–4 was performed except where noted (Supplementary Table 8) and a mean pK$_i$ and SEM was calculated using these replicate experiments. Affinities at off-targets are provided in Supplementary Tables 9, 10.

## Gq-dissociation and β-arrestin2 recruitment BRET assays

To measure 5-HT receptor-mediated β-Arrestin2 recruitment as measured by BRET[1], HEK293T cells (ATCC Cat# CRL-11268; mycoplasma-free) were subcultured in DMEM supplemented with 10% dialyzed FBS (Omega Scientific) and were co-transfected in a 1:15 ratio with human or mouse 5-HT receptors containing C-terminal fused *renilla* luciferase (*R*Luc8), and a Venus-tagged N-terminal β-arrestin2 using 3:1 ratio of TransiT-2020 (Mirus). $5\text{-HT}_{2A}$ receptor mutants were designed and performed using Q5 mutagenesis kit (New England Biolabs). All DNA was sequence verified using Sanger nucleotide sequencing (Retrogen, San Diego, CA). To measure 5-HT receptor-mediated Gq activation via Gq/γ1 dissociation as measured by BRET[2], HEK293T cells were subcultured in DMEM supplemented with 10% dialyzed FBS and were co-transfected in a 1:1:1:1 ratio with *R*Luc8-fused human Gαq (Gαq- *R*Luc8), a GFP[2]-fused to the C-terminus of human Gγ1(Gγ1-GFP[2]), human Gβ1, and 5-HT receptor using TransiT-2020[84]. After at least 18-24 hours, transfected cells were plated in poly-lysine coated 96-well white clear bottom cell culture plates in DMEM containing 1% dialyzed FBS at a density of 25-40,000 cells in 200 μl per well and incubated overnight. The next day, media was decanted and cells were washed with 60 μL of drug buffer (1× HBSS, 20 mM HEPES, pH 7.4), then 60 μL of drug buffer was added per well. Cells were pre-incubated at in a humidified atmosphere at 37 °C before receiving drug stimulation. Drug stimulation utilized 30 μL addition of drug (3X) diluted in McCorvy buffer (1× HBSS, 20 mM HEPES, pH 7.4, supplemented with 0.3% BSA fatty acid free, 0.03% ascorbic acid) and plates were incubated at indicated time and temperatures. Substrate addition occurred 15 minutes before reading and utilized 10 μL of the *R*Luc substrate, either coelenterazine h for β-Arrestin2 recruitment BRET[1] or coelenterazine 400a for Gq dissociation BRET[2] (Prolume/Nanolight, 5 μM final concentration) and was added per well. Plates were read for luminescence at 485 nm and fluorescent eYFP emission at 530 nm for BRET[1] and at 400 nm and fluorescent GFP[2] emission at 510 nm for BRET[2] at 1 second per well using a Mithras LB940 (Berthold) or a PheraStar FSX (BMGLabTech). The BRET ratios of fluorescence/luminescence were calculated per well and were plotted as a function of drug concentration using Graphpad Prism 5 or 9 (Graphpad Software Inc., San Diego, CA). Data were normalized to % 5-HT stimulation and analyzed using nonlinear regression "log(agonist) vs. response" to yield Emax and $EC_{50}$ parameter estimates.

## Calcium flux assays

Stably expressing 5-HT receptor Flp-In 293 T-Rex Tetracycline inducible system (Invitrogen Cat#R78007, mycoplasma-free) were used for calcium flux assays[85]. Cell lines were maintained in DMEM containing 10% FBS, 10 μg/mL Blasticidin (Invivogen), and 100 μg/mL Hygromycin B (GoldBio). Day before the assay, receptor expression was induced with tetracycline (2 μL/mL) and seeded into 384-well poly-L-lysine-coated black plates at a density of 7,500 cells/well in DMEM containing 1% dialyzed FBS. On the day of the assay, the cells were incubated with Fluo-4 Direct dye (Invitrogen, 20 μl/well) for 1 h at 37 °C, which was reconstituted in drug buffer (20 mM HEPES-buffered HBSS, pH 7.4) containing 2.5 mM probenecid. After dye load, cells were allowed to equilibrate to room temperature for 15 minutes, and then placed in a FLIPR[TETRA] fluorescence imaging plate reader (Molecular Devices). Drug dilutions were prepared at 5X final concentration in drug buffer (20 mM HEPES-buffered HBSS, pH 7.4) supplemented with 0.3% BSA fatty-acid free and 0.03% ascorbic acid. Drug dilutions were aliquoted into 384-well plastic plates and placed in the FLIPR[TETRA] (Molecular Devices) for drug stimulation. Fluorescence for the FLIPR[TETRA] were programmed to read baseline fluorescence for 10 s (1 read/s), and afterward 5 μl of drug per well was added and read for a total of 5-10 min (1 read/s). Fluorescence in each well was normalized to the average of the first 10 reads for baseline fluorescence, and then either

maximum-fold peak increase over basal or area under the curve (AUC) was calculated. Either peak or AUC was plotted as a function of drug concentration, and data were normalized to percent 5-HT stimulation. Data was plotted and non-linear regression was performed using "log(agonist) vs. response" in Graphpad Prism 9 to yield Emax and $EC_{50}$ parameter estimates.

## Surface expression/internalization experiments

Surface expression was measured using a HiBit-tagged $5\text{-HT}_{2A}$ receptor and the Nano-Glo HiBit Extracellular Detection System (Promega). N-terminal HiBit-tagged human $5\text{-HT}_{2A}$ receptor was cloned into pcDNA3.1 using Gibson Assembly. HEK293T cells (ATCC Cat#CRL-11268; mycoplasma-free) were transfected into 10-cm tissue culture dishes in a 1:15 ratio of HiBit-tagged human $5\text{-HT}_{2A}$ receptor: human β-Arrestin2 (cDNA Resource Center; www.cDNA.org). Cells were transfected in DMEM 10% dFBS and the next day, cells were plated into either poly-L-lysine-coated 96-well white assay plates (Grenier Bio-One). On the day of the assay, plates were decanted and HEPES-buffered DMEM without phenol-red (Invitrogen) was added per well. Plates were allowed to equilibrate at 37 °C in a humidified incubator before receiving drug stimulation. Compounds (including 5-HT as control) were serially diluted in McCorvy buffer (20 mM HEPES-buffered HBSS, pH 7.4 supplemented with 0.3% BSA fatty-acid free and 0.03% ascorbic acid), and dilutions were added to plates in duplicate (96). Plates were allowed to incubate at 37 °C for 1 hour in a humidified incubator or a specified time point. Approximately 15 minutes before reading, LgBit and coelenterazine h (5 uM final concentration) were added to each well. Plates were sealed to prevent evaporation and read on either a PheraStar FSX (BMB Lab Tech) or Mithras LB940 (Berthold Technologies) at 485 nm at 37 °C for time-capture quantification of internalization or loss of surface expression. Luminescence was plotted as a function of drug concentration using Graphpad Prism 5 or 9 (Graphpad Software Inc., San Diego, CA). Data were analyzed using nonlinear regression "log(agonist) vs. response" to yield Emax and $EC_{50}$ parameter estimates and normalized to % 5-HT surface expression, which a full concentration-response curve was present on every plate.

## Induced fit docking

Docking simulations of six of the 25N ligands, selected to assess the binding modes and provide potential insight into the SAR around transducer or receptor selectivity, were carried out against the active state human $5\text{-HT}_{2A}$ receptor (PDB: 6WHA) using the Induced-Fit Docking (IFD) protocol[86] of the Schrodinger Suite (2020a). IFD was run with extended sampling enabled, using the default setting of residues within 5 Å of the experimental ligand (25N-NBCN) and other parameters maintained at their default values, and ligand structures processed using the LIGPREP tool (target pH of 7) of the Schrödinger (release 2020c).

Poses generated in the IFD runs were analyzed with PyMol visualization application (The PyMOL Molecular Graphics System, Version 2.0 Schrödinger, LLC).

## Estimation of ring substituent effects on molecular electrostatic potential

A series of model compounds were constructed that included only the benzylamine or naphthylamine portion of the corresponding ligand structure. Quantum mechanical optimization of the model compounds, in both neutral and cationic forms, was carried out using density functional theory (DFT) with the Jaguar tool of the Schrodinger suite using the B3LYP functional and 6–311 g**++ basis set, with the exceptions of the models for 25N-NBI, where the halogen atom was not supported, and a smaller basis (3–21 g*++) was substituted. We computed Hirshfeld partial atomic charges (Supplementary Table 4) for the model compounds using the optimized wavefunctions as a proxy

for overall MEP. Hirshfeld charges are computed from a spatial partitioning of the electron density, and have been shown to correlate well with a variety of properties of aromatic molecules[87].

## Molecular dynamics (MD) simulations

Simulations were performed for each of two ligands (25CN-NBOH or 25N-N1-Nap (16) in complex with the human 5-HT$_{2A}$ serotonin receptor in a lipid bilayer of 1,2-dipalmitoyl-sn-glycero-3-phosphocholine (DPPC) and SPC water with enough Cl$^-$ and Na$^+$ ions to neutralize the system at biologic salinity (0.15 M ionic strength). For the 25CN-NBOH system a model of the protein-ligand part of the system was made from chain A of the 6WHA PDB entry, downloaded from the OPM server[88]. Missing extracellular loops were filled in with those from chain A of PDB entry 6WGT. Schrödinger's Protein Preparation Wizard[89] was employed to use Prime[90] to fill in missing side chains, and to use Epik[91] to select the appropriate tautomers and protonation states of the protein and ligand. Due to anticipated difficulty modeling the long ICL3, the helices of TM5 and TM6 were capped, along with the N and C termini, with polar terminating residues: COOH for residue Q262$^{5.66}$ and NH$_2$ for I315$^{6.27}$. During pilot simulations, entanglement of polar sidechains in the membrane occurred. To prevent this, it was found necessary that the rotamers for the loops and capped termini whose sidechains would otherwise interact with the polar heads of the lipid molecules be manually adjusted to ensure adequate water solvation at the beginning of the simulation. This was achieved by maximizing the magnitude of atomic coordinate z of terminal side-chain atoms, where the z-axis is perpendicular to the membrane and $z = 0$ locates the middle of the bilayer. Another issue identified during preliminary simulations was that DPPC tails would intrude into the orthosteric pocket between helices TM4 and TM5, causing the nitrile end of the ligand to project up toward the extracellular side of the pocket. On careful inspection, it was discovered that there is a stable position for a water molecule among residues D120$^{2.50}$, S162$^{3.39}$, and N376$^{7.49}$. Manual introduction of a solvent molecule into this void volume successfully prevented lipid intrusion.

A disulfide bond between C148$^{3.25}$ and C227$^{45.50}$[18] was added to the topology, as well one between C349$^{6.61}$ and C353$^{ECL3}$. Each ligand was protonated at the basic nitrogen (as confirmed by Epik) and topology parameter files needed for subsequent dynamics simulation were determined using the ATB server [https://atb.uq.edu.au/]. Partial atomic charges for the ligands were computed via Schrödinger's Jaguar tool using the density-functional method (B3LYP-D3 functional, 6-31 G** basis), with discrete charges derived from the geometry-optimized wavefunction using the Hirshfeld approach (Supplementary Table 4). The total protein comprised 278 residues with 2849 atoms. The remaining system consisted of 6933 water molecules, 29 Cl$^-$ ions, 20 Na$^+$ ions, and 88 DPPC molecules. 25CN-NBOH has 44 atoms. The total number of atoms in the system is 28,137. The orthorhombic initial system dimensions were 61.319 × 60.911 × 97.412 Å$^3$.

The protein-ligand complex for 25N-N1-Nap (16) starts with the structure from the top-ranked pose determined by induced fit docking (IFD). A chimeric homology model was created in Maestro using the IFD structure as the template everywhere except at the missing loops; these are taken from the 25CN-NBOH-system. This resulting protein-25N-N1-Nap (16) homology model was superposed (using the positions of the TM C$_\alpha$s) onto the protein-ligand complex in the full (water-ion-lipid-protein-ligand) unrelaxed construct built for the 25CN-NBOH simulation. The protein-25CN-NBOH complex was then removed and the 25N-N1-Nap (16)-protein complex substituted in its place. As the IFD docking involved minimal changes to backbone conformation, the resulting model for the 25N-N1-Nap (16) ligand had helix and loop positions nearly identical to those of the 25CN-NBOH model, permitting a direct structural substitution. That said, prior to any further computations, a restrained minimization was carried out with only the membrane/solvent system fully mobile, to remove any high-energy

interatomic overlaps that might have been inadvertently produced by the alignment and substitution. The 25N-N1-Nap (16) simulation comprised a total of 28,143 atoms (50 atoms for the ligand and otherwise identical to the 25CN-NBOH system). The orthorhombic initial system dimensions were 61.864 × 61.453 × 96.12 Å$^3$.

## Simulation protocols and analysis

Both sets of simulations had three phases: minimization (unrestrained steepest decent), equilibration, and a production run. The equilibration phases for both sets comprise a restrained NVT simulation followed by a restrained NPT one. In both, the protein and ligand atoms were restrained close to their initial positions by a harmonic potential with force constant $k = 1000$ kJ/mol/nm (23.9 kcal/mol/Å). Both production simulations were 250-ns NPT unrestrained MD simulations. The force field used was GROMOS96 54a7[92]. The MD engine used was GROMACS v2021.2. The energy parameters in common between the 25CN-NBOH and the 25N-N1-Nap (16) simulations can be found in Supplementary Table 23. The parameters that vary between stages and systems can be found in Supplementary Table 24.

The chi$_2$ dihedral data were measured by GROMACS's chi command every 10 ps. The histogram bin width was one degree. Plots were made in Grace v5.1.25. The time-series heat maps of the pocket-ligand nonbonded interaction energy were made with Seaborn and Matplotlib, and also using custom python code (see Custom Code). The data were calculated by GROMACS's mdrun command. The pocket residues were determined by Schrödinger's Maestro (all residues within 5 Å of the ligand). Protein cartoons and geometric measurements were made in VMD[93] and Maestro. Twelve replicates of the simulations for the two ligands were performed by carrying out the same protocols for system preparation and equilibration, but with assignment of independent random atomic velocities to the starting structures. These simulations provided enhanced sampling of configurations. While transitions between states were observed within some single replicate trajectories, the observed states were closely similar to those found in the primary simulations.

## Animal behavioral experiments

Male C57BL/6 J mice (6–8 weeks old) from Jackson Labs (Bar Harbor, ME, USA) were used for the behavioral experiments. The mice were housed on a reversed light-dark cycle (lights on at 1900 h, off at 0700 h,) in an AALAC-approved vivarium at the University of California San Diego. Mice were housed up to four per cage in a climate-controlled room and with food and water provided ad libitum except during behavioral testing. Testing was performed between 1000 and 1800 h (during the dark phase of the light-dark cycle). The studies were conducted in accordance with National Institutes Health (NIH) guidelines and were approved by the University of California San Diego Institutional Animal Care and Use Committee (Protocol #S17044).

The drug solutions used for the behavioral experiments were prepared as follows: pimavanserin hemitartrate (MedChemExpress), 25C-NBOH hydrochloride, 3,4-dimethoxy-4-methylphenethylamine hydrochloride (desoxy), and 4-cyclopropyl-3,5-dimethoxyphenethylamine hydrochloride (CPM) were dissolved in sterile water; 6-fluoro-N,N-diethyltryptamine (6-F-DET; donated by the Usona Institute, Fitchburg, WI, USA) was dissolved in sterile water acidified with HCl to pH 5; YM-254,890 (FUJIFILM Wako Chemicals USA, Richmond, VA, USA) was dissolved in 100% dimethyl sulfoxide (DMSO); edelfosine (Tocris Bioscience, Minneapolis, MN, USA), ( ± )−2,5-dimethoxy-4-iodoamphetamine hydrochloride ((±)-DOI; Cayman Chemical, Ann Arbor, MI, USA), R-(−)−2,5-dimethoxy-4-iodoamphetamine hydrochloride (R-(−)-DOI; donated by the National Institute on Drug Abuse, Rockville, MD, USA), phencyclidine hydrochloride (PCP; Sigma-Aldrich), N,N-diethyltryptamine fumarate (DET; donated by the National Institute on Drug Abuse, Rockville, MD, USA), 5-methoxy-N,N-dimethyltryptamine hemifumarate (5-MeO-DMT; donated by

the National Institute on Drug Abuse, Rockville, MD, USA), 6-methoxy-*N,N*-dimethyltryptamine hydrochloride (6-MeO-DMT), 25N-NBOMe hydrochloride, and 25N-NB hydrochloride were dissolved in isotonic saline; 25C-NBMD hydrochloride was dissolved in sterile water containing 2% Tween-80 (v/v); 25N-NBOEt hydrochloride, 25N-NB-2-OH-3-Me hydrochloride, 25N-NBPh hydrochloride, 25O-N1-Nap hydrochloride, 2C2-N1-Nap hydrochloride, 25O-NBOMe hydrochloride, 25O-NBPh-10′-OH hydrochloride, 25O-NB-3-I hydrochloride, 25O-NBcP hydrochloride, 2C-N hydrochloride, and M100907 were dissolved in sterile water containing 5% Tween-80 (v/v); 25N-NBNO$_2$ hydrochloride was dissolved in sterile water containing 20% hydroxypropyl-β-cyclodextran (w/v); 25N-NBBr hydrochloride was dissolved in sterile water containing 5% Tween-80 (v/v) and 20% hydroxypropyl-β-cyclodextran (w/v); for the remaining 25 N derivatives, the hydrochloride salts were dissolved in sterile water containing 1% Tween-80 (v/v). 25N derivatives, 25O-NBOMe, 25O-NBPh-10′-OH, 25O-NB-3-I, 25O-NBcP, 25O-N1-Nap, 2C2-NBOMe, 2C2-N1-Nap, and 25D-N1-Nap were injected SC (5 mL/kg or 10 mL/kg) to avoid first-pass effects and ensure stable pharmacokinetics; M100907 was injected SC (5 mL/kg); pimavanserin, edelfosine, *R*-(−)-DOI, (±)-DOI, 25C-NBOH, desoxy, CPM, 25C-NBMD, DET, 6-F-DET, 5-MeO-DMT, 6-MeO-DMT, and PCP were injected IP (5 mL/kg); YM-254,890 was injected into the lateral ventricle (2 μL) over 1 min. Slightly different procedures were used for the for the tolerance experiments (see below). HTR experiments with 1 mg/kg DOI were performed using either the racemate or the *R*-enantiomer as specified.

### Assessment of the head-twitch response

The head-twitch response (HTR) was assessed using a head-mounted neodymium magnet and a magnetometer detection coil[78]. The mice were allowed to recover from the magnet implantation surgeries for at least 1 week prior to behavioral testing. Mice were tested in multiple HTR experiments, with at least 7 days between studies to avoid carryover effects. HTR experiments were conducted in a well-lit room, and the mice were allowed to habituate to the room for at least 1 h prior to testing. Mice were tested in a 12.5-cm diameter glass cylinder surrounded by a magnetometer coil. Coil voltage was low-pass filtered (2 kHz), amplified, and digitized (20-kHz sampling rate) using a Powerlab (model /8SP or 8/35) with LabChart software (ADInstruments, Colorado Springs, CO, USA). Head twitches were identified in the recordings off-line by their waveform characteristics[94] or using artificial intelligence[95]. HTR counts were analyzed using one-way ANOVAs or one-way Welch ANOVAs (in cases where groups showed unequal variances). Tukey's test or Dunnett's T3 multiple comparisons test was used for *post hoc* comparisons. Significance was demonstrated by surpassing an α level of 0.05. Median effective doses (ED$_{50}$ values) and 95% confidence intervals for dose-response experiments were calculated by nonlinear regression (Prism 9.02, GraphPad Software, San Diego, CA, USA).

### Assessment of PCP-induced locomotor activity

The mouse behavioral pattern monitor (BPM) was used to assess locomotor activity[96]. Each mouse BPM chamber (San Diego Instruments, San Diego, CA, USA) is a transparent Plexiglas box with an opaque 30 × 60 cm floor, enclosed in a ventilated isolation box. The position of the mouse in *x,y* coordinates is recorded by a grid of 12 × 24 infrared photobeams located 1 cm above the floor. A second row of 16 photobeams (parallel to the long axis of the chamber, located 2.5 cm above the floor) is used to detect rearing behavior. Holepoking behavior is detected by 11 1.4-cm holes that are situated in the walls (3 holes in each long wall, 2 holes in each short wall) and the floor (3 holes); each hole is equipped with an infrared photobeam. The status of each photobeam is sampled every 55 ms and recorded for offline analysis. Mice were allowed to habituate to the testing room for at least 1 h prior

to testing. For the BPM experiments, mice (*n* = 6/group) were pretreated SC with test drug or vehicle, PCP (5 mg/kg) or vehicle was injected IP 10 minutes later, and then the mice were placed in the BPM chambers 10 minutes after the second injection and activity was recorded for 60 minutes. Locomotor activity was quantified as distance traveled, which was analyzed in 20-minute blocks using a three-way ANOVA, with pretreatment and treatment as between-subject variables and time as a within-subject variable. Tukey's test was used for post hoc comparisons. Significance was demonstrated by surpassing an α level of 0.05.

### Effect of Repeated Treatment with 5-HT$_{2A}$ receptor ligands on Head-Twitch Response

In the first experiment, mice (*n* = 6–7/group, 19 total) were injected SC (5 mL/kg) once daily with vehicle (water containing 5% Tween-80), 25N-N1-Nap hydrochloride (20 mg/kg), or (±)-DOI hydrochloride (10 mg/kg) for five consecutive days. Twenty-four hours later, all of the mice were challenged with an IP injection of 1 mg/kg (±)-DOI hydrochloride and then HTR activity was recorded for 40 minutes. In the second experiment, mice (*n* = 6/group, 12 total) were injected IP (5 mL/kg) once daily with vehicle (water) or pimavanserin hemitartrate (1 mg/kg) for five consecutive days. Twenty-four hours later, all of the mice were challenged with an IP injection of 1 mg/kg (±)-DOI hydrochloride and then HTR activity was recorded for 40 minutes.

### Reporting summary

Further information on research design is available in the Nature Portfolio Reporting Summary linked to this article.

## Data availability

All data generated in this study are included in this article, Supplementary Information, and in the Supplementary Data files. Source data are provided with this paper. PDB deposited structural data are as follows: 6WHA 5-HT$_{2A}$ cryo-EM structure in complex with 25CN-NBOH; 6WGT 5-HT$_{2A}$ crystal structure in complex with LSD; 6A94 5-HT$_{2A}$ crystal structure in complex with zotepine; 6A93 5-HT$_{2A}$ crystal structure in complex with risperidone; 6WH4 5-HT$_{2A}$ crystal structure in complex with methiothepin. Source data are provided with this paper.

## Code availability

Custom code is available at github: https://github.com/jruhym/jNotebooksForMD/blob/trunk/pocket-ligandInteractionTimeSeriesHeatmap.ipynb.

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

## Acknowledgements

This work was supported by Medical College of Wisconsin Research Affairs Counsel Pilot grant (JDM), National Institutes of Health General Medical Sciences grant NIGMS R35GM133421 (JDM) and National Institutes of Health Drug Abuse NIDA R01DA041336 (ALH), as well as by the Veteran's Administration VISN 22 Mental Illness Research, Education, and Clinical Center (ALH). We thank technical assistance from Lisa McNally.

## Author contributions

The study was conceived by J.W., A.L.H. and J.D.M. Compounds were synthesized and analytically characterized by J.W., H.M., J.G. and T.F. BRET, calcium flux, binding, mutagenesis, kinetics, and internalization experiments were performed by A.B.C., M.M.C., J.K.L., H.A.B., E.M.B., J.J.H. and E.I.A. under the supervision of J.D.M. In vivo behavioral experiments were performed by A.L.H., R.K., A.K.K. and B.C. A.M.S. provided compounds and input on the experiments. R.Z. and A.J.H conducted the docking and molecular dynamics simulations and helped write docking and MD sections. J.W., A.L.H. and J.D.M. drafted and edited the original manuscript. All authors reviewed the manuscript.

## Competing interests

JW, ALH, and JDM submitted a patent application for the compounds in this study entitled "Selective, partial, and arrestin-biased 5-HT$_{2A}$ agonists with utility in various disorders," PCT WO2022241006A1. AKK is currently an employee of Gilgamesh Pharmaceuticals. All other authors declare no competing interests.
