## [Peer Review File · Nature Communications]

Reviewers' Comments:

Reviewer #1:

Remarks to the Author:

The manuscript by Wallach et al. is a comprehensive study examining structure function of ligands at serotonin 5-HT_{2A} receptors to attempt to define signaling effector pathways relevant to producing mouse head twitch responses. To do this, a series of N-benzyl derivatives were synthesized that altered electrostatic and electrodensity properties of the N-benzyl moiety that resulted in altering affinities/efficacies/and selectivities at 5-HT_{2A/2B/2C} receptors. Within this strategy molecules were found to be functionally selective for either G_q or Barr2 signaling. A main finding was that the G_q ligand efficacy correlated with HTR magnitude, and efficacy of Barr2 recruitment did not. Further that an efficacy of 70% or greater was required to produce HTR, which allowed prediction of behavioral activity in the HTR based on receptor signaling and G_q efficacy. In silico docking and molecular dynamics were used to identify specific residues in the orthosteric binding pocket responsible for this functional selectivity, and W336 was found to be a toggle switch depending on how it was engaged. Internalization and desensitization studies were performed, as well as behavioral assays that showed the Barr2 biased ligands have antipsychotic like activity in a PCP/locomotor assay. The manuscript is well written, data clearly presented, and conclusions largely supported by the data. Their findings inform on a long-standing question regarding the pathways underlying psychedelic drug behavioral responses, and are of high impact. Further, the new chemicals/tools they have developed should be extremely valuable to the field.

Although the use of BRET-based systems takes receptor density and reserve out of the equation for interpreting results on efficacy and potency, as mentioned in the manuscript, they do not inform on affinity or efficacy bias. Is it possible that in your results from BRET experiments where you show there is no clear biased agonism between G_q and Barr2 amongst different psychedelics at the 5-HT_{2A} receptor, some may actually be either affinity or efficacy biased between these pathways and that these potential biases may have roles in vivo?

The internalization studies compared the Barr2-biased ligands to pimivanserin, which does not induce internalization, and the statement is made that these new ligands are "distinct from 5-HT_{2A} antagonists in their ability to downregulate the 5-HT_{2A} receptor." The way this is worded makes it sound like all 5-HT_{2A} antagonists do not produce internalization, this is incorrect. Several 5-HT_{2A} antagonists produce internalization such as ketanserin and clozapine.

Other signaling pathways have been implicated as potentially relevant downstream of 5-HT_{2A} activation that have shown ligand bias such as arachidonic acid production. Have you looked at AA production, do you think it is relevant for anything?

The data presented suggest that efficacy at G_q is the driving signaling pathway for HTR. These seem to be at odds with the recent publication by Kaplan et al. 2020, where they describe G_q biased agonists with efficacy greater than 70% that have little to no HTR response. How do you reconcile their findings with respect to your data?

Is it possible that the current findings with regard to signaling and behaviors are specific for the N-benzyl class of psychedelic? Might other scaffolds afford different pharmacology and functional selectivity? For example a tryptamine (or ergoline) biased towards Barr2 and away from G_q activating HTR and one the other way around not?

Just because HTR is correlated with aspects of potencies of psychedelics in humans, does it necessarily follow that if a drug does not produce HTR in mice it will have no subjective psychedelic effects in humans?

Minor:

Line 66: Clarify which aspect of human psychedelic activity the HTR correlates with. Intensity? Subjective effects? Potency of intensity?

Line 68: Clarify that there are 15 known serotonin receptors in mammals (14 in humans). Or do you mean to say that LSD has affinity for 12 out of the 15 known mammalian serotonin receptors?

Line 70: The statement is misleading in that it gives the reader the impression that all serotonergic psychedelics have pronounced polypharmacology across many aminergic GPCRs. While this is certainly true for the ergolines, it is not as true for the phenethylamines, which can be very selective for 5-HT₂ receptors. Perhaps rephrase to state that many serotonergic psychedelics, especially of the tryptamine and ergoline classes, lack specificity.

Line 73: The definition of functional selectivity is not quite correct. Rephrase to say that ligands can stabilize certain conformational states of the receptor that each energetically favor coupling to different effectors and downstream signaling pathways.

Supplemental Table 7: Define "ND"

Reviewer #2:

Remarks to the Author:

In this article, the authors include a massive amount of experimental work, particularly BRET-based G protein dissociation and b-arrestin recruitment assays to test the effect of a battery of psychedelic serotonin 5-HT_{2A} agonists in HEK293 cells. They also synthesize new phenethylamine psychedelics, test their function in HEK293 cells using the same experimental system, and evaluate their behavioral effects in rodent models.

The main conclusion of the article is that Gq coupling and not b-arrestin recruitment is necessary for the head-twitch response induced by phenethylamine psychedelics. Although again the amount of work included in this article is impressive (experimental weaknesses particularly *in vitro* are minor as described below), the conclusion proposed by the authors is not new but merely incremental. Thus, it has already been reported that the number of head-twitches induced by the phenethylamine psychedelic DOI is reduced in Gq-KO mice (PMID: 17493641), but the effect of the same psychedelic on HTR is unaffected in b-arrestin2-KO mice (PMID: 18195357, PMID: 35900876). This reduces novelty of the findings included in the current version of this manuscript.

Other comments:

1-Although the experiments *in vitro* related to G protein coupling and b-arrestin recruitment are well designed, the BRET assay by itself does not give information about b-arrestin-dependent function but merely b-arrestin recruitment upon agonist exposure. This limits the functional conclusion that the authors mention in the manuscript.

2-*In vitro* findings are lacking affinity assays for the newly designed 5-HT_{2A} agonists. They tested via BRET assays and other assays including Ca²⁺ release their functional properties, but they did not test affinity targeting the 5-HT_{2A} receptor (via displacement of [³H]ketanserin, [³H]LSD or [³H]5-HT binding, as an example). This is a limitation of the conclusions presented by the authors since affinity and potency are different pharmacological concepts affected differently by the structure of the new ligands.

3-Minor critique related to *in vitro* assays: In Fig 1, it would be interesting to include already available non-psychedelic 5-HT_{2A} agonists such as lisuride or 2-Br-LSD

3- It is not clear why some head-twitch behavior data are presented as HTR counts but other datasets are presented as HTR counts (% of control).

4-The behavioral part of the paper is confusing. The introduction of the article mentions the potential antidepressant effects of new non-hallucinogenic 5-HT_{2A} agonists, but behaviorally the authors tested HTR and its correlation with Gq dissociation and b-arrestin2 recruitment (which once again it is not novel since it has already been reported that HTR induced by phenethylamine psychedelics is not b-arrestin2-dependent).

5-Mechanistically (in vivo in mouse models) how repeated administration of 25N-N1-Nap (a beta-arrestin2-biased compound according to authors' findings) induces tolerance (ie, reduced the ability of DOI to induce HTR) is confusing since previous in vitro assays have reported that agonist-induced internalization of 5-HT2A is beta-arrestin-independent (PMID: 11069907), and in vivo it has been shown that development of tolerance to the effect of DOI on HTR is unaffected in beta-arrestin2-KO mice (PMID: 35900876).

6-Preclinical translational validation of the in vitro assays is weak. As mentioned above, the authors showed interest on antidepressant-like activity of new non-psychedelic 5-HT2A agonists, but then they merely test effect of their new 5-HT2A agonists on PCP-induced hyperlocomotor activity (Fig 6F, G). More elaborated antidepressant-like behavioral or synaptic plasticity effects of their new 5-HT2A agonists are needed to validate the functional relevance of their in vitro findings.

Reviewer #3:

Remarks to the Author:

The John McCorvy team's manuscript addresses the topic of psychedelics, which is considered one of the most important in the field of neuropsychopharmacology. The therapeutic potential of psychedelics known since the 20th century and synthesized in recent years has been intensively evaluated in preclinical studies and clinical trials. A snapshot of the current field of psychedelics shows a shortage of selective 5-HT2A receptor agonists. Similarly, knowledge of functionally selective compounds preferentially engaging selected signaling pathways controlled by the 5-HT2A receptor is even more limited. Therefore, it is very important to provide selective compounds for the 5-HT2A receptor to understand their effects at the molecular level, trying to translate their biological and molecular focus into clinical trials.

The work by J. McCovy team perfectly fits the above context. The manuscript reports on design, synthesis, biological and pharmacological evaluation of a series of phenethylamine derivatives to determine an impact of 5-HT2A receptor signaling pathways to the psychedelic activity.

Compound design was based on literature data supported by molecular docking and molecular dynamics simulations (with interesting structural bias). The authors extended, known from literature, the concept of introducing a benzyl group into an amino group to increase the affinity of the template molecule for 5-HT2 receptors. On the other hand, a number of small design modifications based on rational incentives make this series of compounds very interesting. The value of the work results from the detailed characterization of the biological activity of the compounds - both in receptor and functional assays, including signaling pathways controlled by the 5-HT2A receptor, profiling of the receptor's selectivity towards several GPCRs.

The authors formulated the hypotheses very rationally and consistently presented the biological and behavioral data confirming their achievements: 1) discovery of a potent and selective 5-HT2A receptor agonist with different preferences in Gs and beta-arrestin signaling, 2) discovery of the 5-HT2A receptor, threshold of effectiveness for induction HTR in mice, 3) determining that a beta-arrestin-targeted 5-HT2A receptor agonist produces an antipsychotic-like effect (reflecting the amelioration of the positive properties of schizophrenia).

The results are presented in a clear manner and then reference is made to the current literature. The methods used to draw conclusions are correct. I really like reading the manuscript. It is worth noting that the manuscript has great prospective value.

Major comment:

1. Authors found interesting relationship between in vitro Gs activity and propensity to evoke HTR in mice. Such statement is fine upon consideration that all compounds proceed similar in vivo metabolism. This is important, since no pharmacokinetic data was provided for evaluated compounds. Of note, the effect from HTR may not be a proof for compound brain distribution. How about the brain/plasma preference in vivo....?

In such situation, assessment of compounds' metabolic stability using in vitro methods, should

support whether the effect observed in vivo might be assigned to the parent molecules. Evaluation of metabolic stability at least for compounds tested in vivo would be advantageous.

2. How the effect of compounds at G11 protein may impact the psychedelics effect? Do authors think G11 signaling might be of minor focus in optimization of antidepressant-/antipsychotic-like properties?

3. May authors comment in the manuscript, if the tolerance or tachyphylaxis effects may impact the antipsychotic-like effect of 5-HT2A receptor β -arrestin biased ligands?

4. Comparison of data reported in Figure 4F-4H shows that the most selective compound for 5-HT2A receptor among presented, is the less potent in reversing DOI effect. Authors should comment in the manuscript on compounds selectivity for 5-HT2B receptor and 5-HT2C receptor.

Minor comments:

1. The word "receptor" seems to have been lost in the title of the manuscript. I recommend replacing "5-HT2A" with "5-HT2A receptor" in the title and along the manuscript. The same is true for a "5-HT2A receptor agonist."

2. It seems that it may be more descriptive to characterize the different chemotypes/subseries than to give short acronyms such as scaffold 2C-x (line 340). Such a modification would improve the understanding of structural differences at a glance.

3. Methods - please specify the number of mice used for HTR

4. Figures/Methods - Racemic DOI and R-DOI were used for HTR induction. Please check and disclose why R-DOI was used in selected tests?

5. Figures 4A and 4B-D - please standardize the method of data presentation

6. Figure 4F-H - please modify the captions under the figures so that the DOI group is clearly visible (I assume the "white" group is the DOI)

7. Figure 6 - Why did the HTR test only use the 20-minute recording time for pimavanserin?

8. Characteristics of hydrochloride salts of the reported compounds (supplementary information) - apart from characteristics of the melting point, elemental analysis is used to exclude/determine the amount of water molecules. . NMR spectra confirm "some water". HRMS does not define an "anti" ion to a primary molecule. Please enter data on compounds' elemental analyses (C,H,N) for at least selected important compounds

9. Appendix Table 1 - please give the general structure of the disclosed compounds

Reviewer #4:

Remarks to the Author:

The study is large encompassing an impressive breadth of data spanning structure-based drug design, molecular dynamics, BRET-assays, mutagenesis and in vivo studies for the 5-HT2A receptor. The study presents strong pharmacological data linking the Gq, but not β -arrestin2, pathway to psychedelic potential measured as mouse head-twitch response. This is of high novelty and importance towards the design of safer drugs targeting this receptor.

This study has identified an agonist, 25N-NBI with higher selectivity for the serotonin 5-HT2A receptor than the previously most selective ligand, 25CN-NBOH. Furthermore, the authors identified agonists with signaling bias for β -arrestin2 over Gq. These ligands represent new important tool compounds for the characterization of 5-HT2A-mediated functions as well as therapeutic targeting.

The paper is well written, and conclusions are clearly explained in both text and illustrations. Thus, this reviewer suggests that it would be suitable in Nature Communications after minor revisions.

Major comments

1. The authors recently published work on Br-LSD (<https://doi.org/10.1016/j.celrep.2023.112203>), which is highly G protein biased, that may appear to contradict this paper suggesting that compounds without G protein activity are not hallucinogenic. If this is correct, could the authors please explain this, perhaps in the Discussion.

2. Recent guidelines for the design and reporting of GPCR ligand bias studies (referenced paper by Kolb et al, 2022) distinguish between physiology-bias and pathway-bias, which are measured using an endogenous and pathway-balanced reference ligand, respectively. In Figure 1D, serotonin

appears to be pathway-balanced, as the Gq dissociation and β -arrestin2 recruitment concentration-response curves are near-identical. The authors should state more clearly in the text whether serotonin is pathway-balanced in their assays- and systems (it is not in studies using other assays or systems).

3. As a further note on the guidelines for reporting of GPCR ligand bias, I would please ask that the authors

- a) include an updated supplementary table of "Experimental parameters critical to the unambiguous description of ligand bias" (a template is provided in Table 3).
- b) Deposit their 5-HT_{2A} Gq dissociation and β -arrestin2 recruitment data in a public database, most relevant being the Biased Signaling Atlas (<https://biasedsignalingatlas.org>).

4. Induced fit docking (IFD) is generally considered as a relatively unprecise docking protocol, especially when multiple residues are selected (for Ala mutation in initial docking and subsequent sidechain placement relative ligand). As a minimum, the authors should:

- a) describe in Methods which residues were selected in the induced fit docking.
- b) explain why not some or all compounds could be docked directly into the 25-CN-NBOH bound structural template.
- c) visualize how the docked poses compare to that of 25-CN-NBOH in the experimental structure template.

Minor comments

L32-33: Here we show that 5-HT_{2A} primarily couples to Gq/11 and β -arrestin2 and that prototypical psychedelics do not show a preference for either effector... Given that the couplings have been shown before, and that authors use "confirm" later in the text, please rephrase to:

Here we confirm that 5-HT_{2A} primarily couples to Gq/11 and β -arrestin2 and show that prototypical psychedelics do not show a preference for either effector...

L72-73: The 5-HT_{2A} receptor, like other GPCRs, can exhibit functional selectivity or biased agonism, which is the ability to stabilize subsets of its downstream effectors, such as Gq/11 coupling and β -arrestin2 recruitment.

This is not a correct definition of biased agonism. Furthermore, biased agonism typically only refers to the ligand whereas the term functional selectivity encompasses both the ligand and system. Authors should use other definition(s). This can be found in the guideline paper or other authoritative reviews.

L75-77: For example, the G protein-biased μ -opioid receptor (MOR)-selective agonist oliceridine was recently approved in the US and is claimed to have an improved tolerability profile (Gan and Wase, 2020).

This claim should never be mentioned without the strong contradictive evidence (e.g., Gillis, A. et al. *Sci. Signal.* 13, eaaz3140 (2020)). The vast majority of the GPCR community no longer believes that signaling bias underlies the potential therapeutic benefits of Oliceridine. The sentence could also be removed altogether.

L229-231: This W6.48 residue, sometimes termed the "toggle switch", has been implicated in GPCR activation and signaling bias and distinct W6.48 rotamer conformations occur in activated and non-activated GPCR structures (Chen et al., 2022; Piekielna-Ciesielska et al., 2020; Suomivuori et al., 2020)

W6.48 does not undergo a sidechain rotamer shift upon activation in most class A GPCRs but is instead part of a TM6 helix backbone rotation (see e.g., <https://doi.org/10.1038/s41594-021-00674-7>). Thus, this should no longer be presented as a general activation switch, although in individual GPCRs or active states differences can occur. This should be clarified to make a correct statement.

Furthermore, did the authors see differences in the rotation of TM6 and W6.48 across different agonist-bound conformations in their MD analysis? If so, please include that information too.

L469-472: For example, influence of the W6.48 toggle switch on the PIF motif and corresponding changes in TM6 orientation were observed in MD studies with β 2-adrenergic receptor biased ligands (Chen et al., 2022), rhodopsin (Zhou et al., 2017), MOR (Piekielna-Ciesielska et al., 2020), and S1PR1 (Xu et al., 2022).

The PIF motif is actually PIW in the majority of class A GPCRs (<https://doi.org/10.1038/s41594-021-00674-7>). This is actually a previous report of the interaction between W6.48 and the previous PIF motif.

REVIEWER COMMENTS

Reviewer #1 (Remarks to the Author):

The manuscript by Wallach et al. is a comprehensive study examining structure function of ligands at serotonin 5-HT_{2A} receptors to attempt to define signaling effector pathways relevant to producing mouse head twitch responses. To do this, a series of N-benzyl derivatives were synthesized that altered electrostatic and electrodensity properties of the N-benzyl moiety that resulted in altering affinities/efficacies/and selectivities at 5-HT_{2A/2B/2C} receptors. Within this strategy molecules were found to be functionally selective for either Gq or Barr2 signaling. A main finding was that the Gq ligand efficacy correlated with HTR magnitude, and efficacy of Barr2 recruitment did not. Further that an efficacy of 70% or greater was required to produce HTR, which allowed prediction of behavioral activity in the HTR based on receptor signaling and Gq efficacy. In silico docking and molecular dynamics were used to identify specific residues in the orthosteric binding pocket responsible for this functional selectivity, and W336 was found to be a toggle switch depending on how it was engaged. Internalization and desensitization studies were performed, as well as behavioral assays that showed the Barr2 biased ligands have antipsychotic like activity in a PCP/locomotor assay. The manuscript is well written, data clearly presented, and conclusions largely supported by the data. Their findings inform on a long-standing question regarding the pathways underlying psychedelic drug behavioral responses, and are of high impact. Further, the new chemicals/tools they have developed should be extremely valuable to the field.

We thank the reviewer for their enthusiasm and value of our chemical tools in this study.

Although the use of BRET-based systems takes receptor density and reserve out of the equation for interpreting results on efficacy and potency, as mentioned in the manuscript, they do not inform on affinity or efficacy bias. Is it possible that in your results from BRET experiments where you show there is no clear biased agonism between Gq and Barr2 amongst different psychedelics at the 5-HT_{2A} receptor, some may actually be either affinity or efficacy biased between these pathways and that these potential biases may have roles in vivo?

We thank the reviewer for this comment. The reviewer is correct in their assessment in the use of the BRET-based system is not sensitive to receptor reserve and other system-dependent artifacts in the interpretation of ligand-dependent bias.

Regarding the point whether there is affinity bias: we did not detect substantial differences comparing Gq and β -arr2 potencies. In fact, part of the reasoning of using the BRET system is to accurately assess the agonist affinity (K_a) for each of the measured transducers, as opposed to amplified second messenger assays that are receptor expression-dependent. In our previously published study on Ariadne and analogs (PMID: **36521179**), we show that the 5-HT_{2A} BRET EC₅₀s are similar to the affinity (K_i) obtained with the agonist radioligand, I125-DOI. This further supports that EC₅₀s in the BRET assay are in line with agonist affinities obtained in binding studies using an agonist radioligand.

Regarding the point whether there is efficacy bias: we also did not detect substantial differences in Emax comparing Gq and β -arr2, but we do note some exceptions: psilocin, DMT and 2C-I were slightly higher for Gq efficacy compared to β -arrestin2. Because some in vitro assays can inherently produce a greater signal-to-noise and thus “skew” the interpretation of ligand-dependent bias toward the effect with greater signal-to-noise, we specifically show in Fig 1C that both Gq dissociation and β -arr2 signal-to-noise windows are similar as measured by net BRETs. Therefore we are confident that we do not detect substantial efficacy-driven bias for the tested psychedelics when comparing these two assays and under these conditions.

Finally, we also provide these BRET measurements in a kinetic context showing the differences across several time points in Supplementary Fig 1, as ligand-directed biased agonism can also be dynamic through time.

The internalization studies compared the β -arr2-biased ligands to pimivanserin, which does not induce internalization, and the statement is made that these new ligands are “distinct from 5-HT_{2A} antagonists in their ability to downregulate the 5-HT_{2A} receptor.” The way this is worded makes it sound like all 5-HT_{2A} antagonists do not produce internalization, this is incorrect. Several 5-HT_{2A} antagonists produce internalization such as ketanserin and clozapine.

Thank you for this point. We have specified and re-worded as “...distinct from pimavanserin in their ability to downregulate the 5-HT_{2A}R.” We also cite published work on internalization with other 5-HT_{2A} antagonists.

Other signaling pathways have been implicated as potentially relevant downstream of 5-HT_{2A} activation that have shown ligand bias such as arachidonic acid production. Have you looked at AA production, do you think it is relevant for anything?

We are not quite sure of the relevance of the arachidonic acid production to psychedelic potential, but this would be interesting to follow-up. In our study, we specifically focused on the transducers known to proximally engage 5-HT_{2A}, and focused our analysis on G proteins and β -arrestins. Our study does not preclude the relevance of arachidonic acid or other second messengers or signaling cascades that result from these transducers upon activation.

The data presented suggest that efficacy at Gq is the driving signaling pathway for HTR. These seem to be at odds with the recent publication by Kaplan et al. 2020, where they describe Gq biased agonists with efficacy greater than 70% that have little to no HTR response. How do you reconcile their findings with respect to your data?

Thank you for pointing this out. We note that in the Kaplan et al. 2022 study, the compounds were not extremely biased as they still retain significant activity (efficacy and potency) in both Gq and β -arr2 BRET assays as seen in Ext Data Fig 3. Also we note the authors only tested their compounds at two doses (1 and 3 mg/kg), which is not a wide-enough dose range to detect the HTR induced by many 5-HT_{2A} agonists, especially considering their compounds are not very potent in either Gq or β -arr2 BRET assays. Some 5-HT_{2A} agonists only induce the HTR if they are tested at doses >3 mg/kg. For example, in the present investigation, if we had only

tested our compounds at 1 and 3 mg/kg then we would have concluded that 25N-NBBr, 25N-NBI, and 25N-NBOCF2 are inactive in the HTR assay. Inspection of Figure 5 in Kaplan et al 2022 indicates that administration of their (R)-69 compound at 3 mg/kg did increase the number of HTR counts compared to the control group, so the outcome of their experiment may have been different if they had tested the compounds at higher doses. In addition, it is also important to consider that HTR dose-response curves are biphasic, so if the doses used in experiments are too high then activity may be missed. For example, the HTR induced by LSD peaks at 0.2 mg/kg and LSD does not induce the HTR when administered at 1 or 3 mg/kg. Hence the doses tested by Kaplan et al may simply have been too low or too high to detect activity in the HTR assay.

Is it possible that the current findings with regard to signaling and behaviors are specific for the N-benzyl class of psychedelic? Might other scaffolds afford different pharmacology and functional selectivity? For example a tryptamine (or ergoline) biased towards Barr2 and away from Gq activating HTR and one the other way around not?

Thank you for this suggestion. In our revision, we added a new figure to the manuscript (Fig 6), which shows that the activity of psychedelic versus non-psychedelic tryptamines and ergolines is also governed by an efficacy threshold.

Just because HTR is correlated with aspects of potencies of psychedelics in humans, does it necessarily follow that if a drug does not produce HTR in mice it will have no subjective psychedelic effects in humans?

This is a great question but it is not a question our study intended to address fully. That being said, the cross-species potency correlations we report are useful for our determination of “psychedelic potential” but by no means are we intending to address a full spectrum of qualitative effects experienced by humans. The strength of our study is that the HTR is highly predictive of whether a 5-HT_{2A} agonist will exhibit LSD-like psychopharmacology (which we define as psychedelic potential) and we have tested hundreds of 5-HT_{2A} agonists in the HTR assay and we conclude that agonists that produce hallucinogenic effects (LSD-like subjective phenomenology in humans and/or LSD-like discriminative stimulus effects in rats) invariably induce head twitches in male C57BL/6J mice, whereas agonists lacking hallucinogenic effects do not induce the HTR. For example, as we now show in Figure 6, the non-hallucinogenic 5-HT_{2A} agonists lisuride, 2-Br-LSD, 6-F-DET, and 6-MeO-DMT did not induce the HTR in our experiments. Based on those results, drugs that do not produce the HTR in mice would not be predicted to induce psychedelic subjective effects in humans, but it is merely a prediction. Ultimately, non-psychedelic agonists testing in humans will fully validate the qualitative psychedelic effects.

Minor:

Line 66: Clarify which aspect of human psychedelic activity the HTR correlates with. Intensity? Subjective effects? Potency of intensity?

We have reworded the sentence in the Introduction to say that the HTR “predicts human psychedelic activity”. In other words, if a 5-HT_{2A} agonist induces the HTR in mice then it is likely to produce psychedelic effects in humans.

Line 68: Clarify that there are 15 known serotonin receptors in mammals (14 in humans). Or do you mean to say that LSD has affinity for 12 out of the 15 known mammalian serotonin receptors?

Due to word constraints we have cut this sentence and section.

Line 70: The statement is misleading in that it gives the reader the impression that all serotonergic psychedelics have pronounced polypharmacology across many aminergic GPCRs. While this is certainly true for the ergolines, it is not as true for the phenethylamines, which can be very selective for 5-HT₂ receptors. Perhaps rephrase to state that many serotonergic psychedelics, especially of the tryptamine and ergoline classes, lack specificity.

Due to word constraints we have cut this sentence and section.

Line 73: The definition of functional selectivity is not quite correct. Rephrase to say that ligands can stabilize certain conformational states of the receptor that each energetically favor coupling to different effectors and downstream signaling pathways.

We have re-phrased as suggested.

Supplemental Table 7: Define “ND”

We have denoted as “ND = not determined” in the caption.

Reviewer #2 (Remarks to the Author):

In this article, the authors include a massive amount of experimental work, particularly BRET-based G protein dissociation and b-arrestin recruitment assays to test the effect of a battery of psychedelic serotonin 5-HT_{2A} agonists in HEK293 cells. They also synthesize new phenethylamine psychedelics, test their function in HEK293 cells using the same experimental system, and evaluate their behavioral effects in rodent models.

The main conclusion of the article is that Gq coupling and not b-arrestin recruitment is necessary for the head-twitch response induced by phenethylamine psychedelics. Although again the amount of work included in this article is impressive (experimental weaknesses

particularly in vitro are minor as described below), the conclusion proposed by the authors is not new but merely incremental. Thus, it has already been reported that the number of head-twitches induced by the phenethylamine psychedelic DOI is reduced in Gq-KO mice (PMID: 17493641), but the effect of the same psychedelic on HTR is unaffected in β -arrestin2-KO mice (PMID: 18195357, PMID: 35900876). This reduces novelty of the findings included in the current version of this manuscript.

Thank you for your evaluation of our study. We note that in prior studies the conclusions are still conflicting on whether 5-HT_{2A} Gq or arrestin is necessary for psychedelic potential (compare studies PMIDs: 17493641 to 34480046). The prior studies only addressed this conclusion with limited psychedelic agents and global constitutive knock-out mice, and did not test several chemical classes of psychedelic agents and compared them to non-psychedelic agents. Moreover, the over-reliance on the use of global β -arrestin2 knockout mice has added to a similar controversial issue regarding the mu-opioid receptor (MOR) on whether respiratory depression is completely β -arrestin2 dependent.

In our revision, we have highlighted the existence of a 5-HT_{2A}-Gq efficacy threshold to induce the HTR (predictive of psychedelic effects), which is a new finding that has not been proposed and validated as in our study. In our study we use a range of ligands, some that are 5-HT_{2A}-selective and 5-HT_{2A}- β -arrestin biased, and some that are non-psychedelic but still demonstrate 5-HT_{2A} agonism. We leverage these chemical tools to support that a 5-HT_{2A} Gq efficacy threshold is a novel explanation of why certain 5-HT_{2A} agonists such as lisuride, 2-Br-LSD, 6-F-DET, and 6-MeO-DMT are non-hallucinogenic, which are now included in Figure 6 in our revision. The discovery of a 5-HT_{2A}-Gq efficacy threshold also provides a potential strategy that can be used to design new lisuride-like “psychoplastogen” molecules that mimic the therapeutic activity of psychedelics but produce little or no hallucinogenic effects. Furthermore, our study is novel in that this is the first characterization of β -arrestin-biased 5-HT_{2A} agonists devoid of psychedelic potential, which we show may have utility as antipsychotic agents.

Other comments:

1-Although the experiments in vitro related to G protein coupling and β -arrestin recruitment are well designed, the BRET assay by itself does not give information about β -arrestin-dependent function but merely β -arrestin recruitment upon agonist exposure. This limits the functional conclusion that the authors mention in the manuscript.

Thank you for this comment. We have performed in vitro internalization assays and in vivo tolerance studies, which address two potential functional consequences of 5-HT_{2A} β -arrestin2 recruitment and are included in Figure 7.

2-In vitro findings are lacking affinity assays for the newly designed 5-HT_{2A} agonists. They tested via BRET assays and other assays including Ca²⁺ release their functional properties, but they did not test affinity targeting the 5-HT_{2A} receptor (via displacement of [3H]ketanserin, [3H]LSD or [3H]5-HT binding, as an example). This is a limitation of the conclusions presented

by the authors since affinity and potency are different pharmacological concepts affected differently by the structure of the new ligands.

Thank you for the suggestion. In the results, we have included affinities for the compounds in the study and those data are found in Supplementary Tables 8-10.

3-Minor critique related to in vitro assays: In Fig 1, it would be interesting to include already available non-psychedelic 5-HT_{2A} agonists such as lisuride or 2-Br-LSD

Thank you for this suggestion. We have now included these non-psychedelic 5-HT_{2A} agonists and their BRET and HTR data in Figure 6. We also cite some of our recently published data for 2-Br-LSD (Lewis et al., 2023).

3- It is not clear why some head-twitch behavior data are presented as HTR counts but other datasets are presented as HTR counts (% of control).

Thank you for this comment. We now express HTR as total counts where possible throughout the manuscript, except in the case of tolerance experiments in Figure 7. There, we present the normalized results to account for experimental design of comparing to vehicle treated mice in two separate experiments examining 5-HT_{2A} ligand-induced tolerance.

4-The behavioral part of the paper is confusing. The introduction of the article mentions the potential antidepressant effects of new non-hallucinogenic 5-HT_{2A} agonists, but behaviorally the authors tested HTR and its correlation with Gq dissociation and b-arrestin2 recruitment (which once again it is not novel since it has already been reported that HTR induced by phenethylamine psychedelics is not b-arrestin2-dependent).

We apologize for the confusion. We have only cited prior work in the Introduction pertaining to the antidepressant potential for non-hallucinogenic 5-HT_{2A} agonists to explain the context for our investigation, but our results are not intended to directly address that question. Regarding novelty of phenethylamines and their HTR, we now include results with other 5-HT_{2A} agonists from the tryptamine and ergoline classes in Figure 6.

5-Mechanistically (in vivo in mouse models) how repeated administration of 25N-N1-Nap (a beta-arrestin2-biased compound according to authors' findings) induces tolerance (ie, reduced the ability of DOI to induce HTR) is confusing since previous in vitro assays have reported that agonist-induced internalization of 5-HT_{2A} is beta-arrestin-independent (PMID: 11069907), and in vivo it has been shown that development of tolerance to the effect of DOI on HTR is unaffected in b-arrestin2-KO mice (PMID: 35900876).

Thank you for this comment. In the Bhatnagar et al. 2001 study, it was found that both Arr-2 (β -arr1) and Arr-3 (β -arr2) did indeed cause redistribution of 5-HT_{2A} into distinct intracellular compartments. Our study, however, did not address 5-HT_{2A} sorting into those intracellular compartments, but instead was designed to determine if our discovered β -arrestin-biased

agonists could induce loss of surface expression similar to β -arrestin2 recruitment activity as measured using BRET. Likely the comparison between that previous study and our own reflects system-specific parameters and differences. Regarding the study in β -arr2 KO mice, this likely reflects global knock-out artifacts with these mice where developmental adaptations would likely occur to compensate for the loss of arrestin-dependent functions (e.g. compensation by β -arrestin1 is a possibility), which we detail in the Discussion. The role of 5-HT_{2A}- β -arrestin interactions and concomitant trafficking of the receptor needs to be evaluated using a different approach, which is beyond the scope of the purpose of our study.

6-Preclinical translational validation of the in vitro assays is weak. As mentioned above, the authors showed interest on antidepressant-like activity of new non-psychedelic 5-HT_{2A} agonists, but then they merely test effect of their new 5-HT_{2A} agonists on PCP-induced hyperlocomotor activity (Fig 6F, G). More elaborated antidepressant-like behavioral or synaptic plasticity effects of their new 5-HT_{2A} agonists are needed to validate the functional relevance of their in vitro findings.

Similar to the point above, our study did not intend to address the antidepressant potential of non-psychedelic 5-HT_{2A} agonists. Instead, we performed the PCP-induced hyperlocomotor activity assay to determine if β -arrestin-biased 5-HT_{2A} agonists were distinct from known 5-HT_{2A} antagonists. To discuss whether additional preclinical evidence would be needed at this stage or argue this would be better suited for a more long-term study of the utility of β -arrestin-biased ligands. We have clarified this rationale in the results section.

Reviewer #3 (Remarks to the Author):

The John McCovry team's manuscript addresses the topic of psychedelics, which is considered one of the most important in the field of neuropsychopharmacology. The therapeutic potential of psychedelics known since the 20th century and synthesized in recent years has been intensively evaluated in preclinical studies and clinical trials. A snapshot of the current field of psychedelics shows a shortage of selective 5-HT_{2A} receptor agonists. Similarly, knowledge of functionally selective compounds preferentially engaging selected signaling pathways controlled by the 5-HT_{2A} receptor is even more limited. Therefore, it is very important to provide selective compounds for the 5-HT_{2A} receptor to understand their effects at the molecular level, trying to translate their biological and molecular focus into clinical trials.

The work by J. McCovry team perfectly fits the above context. The manuscript reports on design, synthesis, biological and pharmacological evaluation of a series of phenethylamine derivatives to determine an impact of 5-HT_{2A} receptor signaling pathways to the psychedelic activity.

Compound design was based on literature data supported by molecular docking and molecular dynamics simulations (with interesting structural bias). The authors extended, known from literature, the concept of introducing a benzyl group into an amino group to increase the affinity of the template molecule for 5-HT₂ receptors. On the other hand, a number of small design

modifications based on rational incentives make this series of compounds very interesting. The value of the work results from the detailed characterization of the biological activity of the compounds - both in receptor and functional assays, including signaling pathways controlled by the 5-HT_{2A} receptor, profiling of the receptor's selectivity towards several GPCRs.

The authors formulated the hypotheses very rationally and consistently presented the biological and behavioral data confirming their achievements: 1) discovery of a potent and selective 5-HT_{2A} receptor agonist with different preferences in Gs and b-arrestin signaling, 2) discovery of the 5-HT_{2A} receptor, threshold of effectiveness for induction HTR in mice, 3) determining that a beta-arrestin-targeted 5-HT_{2A} receptor agonist produces an antipsychotic-like effect (reflecting the amelioration of the positive properties of schizophrenia).

The results are presented in a clear manner and then reference is made to the current literature. The methods used to draw conclusions are correct. I really like reading the manuscript. It is worth noting that the manuscript has great prospective value.

We thank the reviewer for their enthusiasm and praise of our study.

Major comment:

1. Authors found interesting relationship between in vitro Gs activity and propensity to evoke HTR in mice. Such statement is fine upon consideration that all compounds proceeds similar in vivo metabolism. This is important, since no pharmacokinetic data was provided for evaluated compounds. Of note, the effect from HTR may not be a proof for compound brain distribution. How about the brain/plasma preference in vivo....?

In such situation, assessment of compounds' metabolic stability using in vitro methods, should support whether the effect observed in vivo might be assigned to the parent molecules. Evaluation of metabolic stability at least for compounds tested in vivo would be advantageous.

We thank the reviewer for this comment. There is considerable evidence that brain 5-HT_{2A} occupation is necessary for the HTR. Although serotonin does not induce the HTR in mice after subcutaneous administration (PMID 5302272), it does induce head twitches when administered ICV (PMID 6125909, 20926677). Likewise, while numerous brain-penetrant 5-HT_{2A} antagonists will block the HTR, the peripheral 5-HT_{2A} antagonist xylamidine is ineffective (PMID 6247593). Those findings demonstrate that the HTR is mediated by central and not peripheral 5-HT_{2A} receptors. Furthermore, administration of the 5-HT_{2A} agonist DOI directly into the prefrontal cortex induces the HTR (PMID 9262333). Although 5-HT_{2A} knockout mice do not emit head twitches in response to DOI, the behavior can be rescued if the 5-HT_{2A} receptor is restored to the frontal cortex (PMID 17270739).

The reviewer is correct that correlation analyses based on drug potency can potentially be affected by pharmacokinetic (PK) factors, which is why we present correlation analysis using compounds from respective core chemical scaffolds (25N versus phenethylamines), which reduces the risk that PK differences will derail the analysis. Previous studies have successfully used a similar approach to confirm that the behavioral potencies of psychedelic drugs are correlated with their pharmacological properties at 5-HT_{2A} such as binding affinity (PMID 6513725, 6776558, 2505289, 3127847).

By contrast, our correlation analyses in Fig. 5 and Fig. 6 are based on drug efficacy (max HTR counts), which is independent of drug potency and is highly insensitive to pharmacokinetic factors. As an example, the intrinsic clearance rate for 25I-NBOMe (4.1 L/kg/h) is 20-fold higher than 2C-I (0.20 L/kg/h) (PMID 2451954), so if there had been a corresponding reduction in the magnitude of the HTR induced by 25I-NBOMe due to its high clearance rate then it would probably have been inactive in the paradigm. But it is not difficult to detect the activity of 25I-NBOMe in HTR experiments (PMID 24012658). Our HTR experiments are designed to account for differences in drug PK because where we tailor the dose range used in each experiment so it matches the potency of the compound being tested. If a particular 5-HT_{2A} agonist exhibits poor PK factors (distribution, bioavailability, brain/plasma ratio) then that reduces the potency of the compound in HTR and we compensate by just testing higher doses until we have examined a dose range high enough to capture the full dose-response function, including the maximally effective dose of that particular compound. We know this approach is effective because we have confirmed in several regression analyses performed with multiple chemical scaffolds that max HTR counts are significantly and robustly correlated with measures of 5-HT_{2A}-Gq efficacy, which strongly indicates that our HTR data collection is not being derailed by PK differences. Finally, our experiments in Fig. 4 and Supplementary Fig. 9 with blockade of the HTR with the inactive β -arrestin biased analogs further demonstrate brain penetrance.

2. How the effect of compounds at G11 protein may impact the psychedelics effect? Do authors think G11 signaling might be of minor focus in optimization of antidepressant-/antipsychotic-like properties?

We thank the reviewer for this interesting question. First, we show in our revision that the G11 activity is also dampened with our β -arrestin-biased compounds (Supplementary Fig. 6). Therefore, because the β -arrestin-biased agonists showed an antipsychotic-like profile, we can conclude that G11 signaling may not be an important optimization toward this profile. Further studies are required to determine degree of antagonism of Gq/11 and agonism of β -arrestin2 pathways that contribute to a superior antipsychotic profile.

3. May authors comment in the manuscript, if the tolerance or tachyphylaxis effects may impact the antipsychotic-like effect of 5-HT_{2A} receptor β -arrestin biased ligands?

Thank you for this comment. We did not address if tolerance or tachyphylaxis contributes to the antipsychotic-like profile, and this result only pertains to the HTR. However, we may speculate that tolerance may effectively be what is driving the antipsychotic effects longer-term and thus future study are required to compare the longer-lasting effects of 5-HT_{2A}- β -arrestin-biased agonists compared to traditional 5-HT_{2A}-selective antagonists/inverse agonists.

4. Comparison of data reported in Figure 4F-4H shows that the most selective compound for 5-HT_{2A} receptor among presented, is the less potent in reversing DOI effect. Authors should comment in the manuscript on compounds selectivity for 5-HT_{2B} receptor and 5-HT_{2C} receptor.

Thank you for this point. We have addressed this potency discrepancy in the revision.

We state: “25N-NBPh (17) had lower potency than 25N-N1-Nap (16) in the blockade experiments, which we believe is due to pharmacokinetic differences limiting its CNS distribution, potentially reflecting its higher cLogP (4.8 vs. 4.5, respectively). We thus focused on 25N-N1-Nap (16) in subsequent experiments.”

Furthermore, based on binding data, 25N-NBPh is highly selective for 5-HT_{2A} (pKi = 9.48) vs 5-HT_{2B} (pKi = 5.85) and 5-HT_{2C} (pKi = 6.43). Therefore, it is unlikely that 5-HT_{2B} or 5-HT_{2C} could be involved in the blockade induced by 25N-NBPh. By contrast, 25N-N1-Nap and 25N-NB-2-OH-3-Me have similar affinities for 5-HT₂ subtypes. Many studies have found that 5-HT_{2B/2C} antagonists do not attenuate the HTR induced by DOI (Kennett et al., 1994; Schreiber et al., 1995; Wettstein et al., 1999; Vickers et al., 2001; Fantegrossi et al., 2010, 2014). Therefore, it seems unlikely that the blockade could be a consequence of interactions with 5-HT_{2B} or 5-HT_{2C}.

Minor comments:

1. The word "receptor" seems to have been lost in the title of the manuscript. I recommend replacing "5-HT2A" with "5-HT2A receptor" in the title and along the manuscript. The same is true for a "5-HT2A receptor agonist."

The revised manuscript now abbreviates the “5-HT_{2A} receptor” as “5-HT_{2A}R.” We have added that notation throughout the manuscript as suggested.

2. It seems that it may be more descriptive to characterize the different chemotypes/subseries than to give short acronyms such as scaffold 2C-x (line 340). Such a modification would improve the understanding of structural differences at a glance.

Thank you for the suggestion. We now avoid using short acronyms whenever possible. The one exception is that derivatives of 25N or 2C-N are still described as being members of the “25N series”, but we believe that use is justified and is unlikely to confuse readers because we provide considerable explanation in the text and figures. Moreover, we now provide structures for all compounds in the supplement and in the figures.

3. Methods - please specify the number of mice used for HTR

Thank you for the suggestion. We have modified the bar graphs so they now show the response of the individual mice, which clearly conveys exactly how many mice were used for each group. The Supplemental Tables also show the number of mice used for HTR. In addition, the manuscript now explains the rationale for using male mice in our experiments.

4. Figures/Methods - Racemic DOI and R-DOI were used for HTR induction. Please check and disclose why R-DOI was used in selected tests?

Thank you for this comment. We have edited the manuscript so the text and figures show more clearly whether racemic DOI or R-DOI was used for each experiment. Over the course of this study (5+ years), it has occasionally been difficult to obtain R-DOI, so racemic DOI had to be used for some blockade experiments. However, as we now note in the revised manuscript, we have confirmed that the magnitude of the HTR induced by 1 mg/kg R-DOI and 1 mg/kg racemic DOI is roughly equivalent, hence the substitution of one form of DOI for another should not have any effect on the outcome of the HTR experiments. The goal of the blockade experiments was to determine if the test compound can attenuate the response to a 5-HT_{2A} agonist, so theoretically any 5-HT₂-selective agonist like DOI could be used for the blockade experiments and there is no reason why the same agonist or the same form of DOI has to be used for every experiment, although we have tried to use a uniform experimental design as much as possible.

5. Figures 4A and 4B-D - please standardize the method of data presentation

Thank you for this comment. We have standardized the method of data presentation in Figure 4. We now use bar graphs consistently.

6. Figure 4F-H - please modify the captions under the figures so that the DOI group is clearly visible (I assume the "white" group is the DOI)

Thank you for this comment. We have added a line and a label to the figure showing that all the mice received DOI.

7. Figure 6 - Why did the HTR test only use the 20-minute recording time for pimavanserin?

Thank you for this comment. The design of the pimavanserin blockade experiment matches the design used in Fig 4F-H. For blockade studies with DOI, we often focus on the first 20 minutes because that is the time period during which the response to DOI peaks. Extending the recording time would not have changed the outcome of the experiments.

8. Characteristics of hydrochloride salts of the reported compounds (supplementary information) - apart from characteristics of the melting point, elemental analysis is used to exclude/determine the amount of water molecules. NMR spectra confirm "some water". HRMS does not define an "anti" ion to a primary molecule. Please enter data on compounds' elemental analyses (C,H,N) for at least selected important compounds

Thank you for this suggestion. Language around the analytical characterization of the compounds was clarified in the supplemental section. Anhydrous d₆-DMSO was used as the NMR solvent though in some cases still contained small amounts of water (tends to increase with time even with careful storage). On the 25N compounds, solvent "blanks" were run along with the sample allowing us to subtract the water signal and thus identify any potential hydrates (none were found - although small amounts of water were sometimes present). This has been clarified in the supplemental section. C, H, N elemental analysis has been included for select

compounds; 25N-NBOH (3), 25N-NBOMe (4), 25N-NBMe (6), 25N-NBF (7), 25N-NBCl (8), and 25N-NBOCF₂H (11). HRMS, NMR, and HPLC are provided to support identity and purity and we did not obtain elemental analysis on all compounds but do provide HPLC, ¹H, ¹³C, (¹⁹F where relevant) and ²D NMR, and HRMS.

9. Appendix Table 1 - please give the general structure of the disclosed compounds

Thank you for this suggestion. In our revision, Supplemental Figures 2 contains structures of the novel compounds, Supplemental Figure 11 contains structures of the examined psychedelics, and we have included chemical structures of all other compounds throughout the figures.

Reviewer #4 (Remarks to the Author):

The study is large encompassing an impressive breadth of data spanning structure-based drug design, molecular dynamics, BRET-assays, mutagenesis and in vivo studies for the 5-HT_{2A} receptor. The study presents strong pharmacological data linking the G_q, but not β -arrestin₂, pathway to psychedelic potential measured as mouse head-twitch response. This is of high novelty and importance towards the design of safer drugs targeting this receptor.

This study has identified an agonist, 25N-NBI with higher selectivity for the serotonin 5-HT_{2A} receptor than the previously most selective ligand, 25CN-NBOH. Furthermore, the authors identified agonists with signaling bias for β -arrestin₂ over G_q. These ligands represent new important tool compounds for the characterization of 5-HT_{2A}-mediated functions as well as therapeutic targeting.

The paper is well written, and conclusions are clearly explained in both text and illustrations. Thus, this reviewer suggests that it would be suitable in Nature Communications after minor revisions.

Thank you for this assessment of the manuscript.

Major comments

1. The authors recently published work on Br-LSD (<https://doi.org/10.1016/j.celrep.2023.112203>), which is highly G protein biased, that may appear to contradict this paper suggesting that compounds without G protein activity are not hallucinogenic. If this is correct, could the authors please explain this, perhaps in the Discussion.

Thank you for pointing this out. In our revision, we have added a new Figure 6, which shows that 5-HT_{2A}-G_q efficacy of 2-Br-LSD falls below the 70% E_{max} threshold required to induce the HTR and hallucinogenic effects. Hence, the 5-HT_{2A} activity of 2-Br-LSD is consistent with the results of the present investigation and it makes sense that 2-Br-LSD is non-hallucinogenic given that its G_q E_{max} is about 60%.

2. Recent guidelines for the design and reporting of GPCR ligand bias studies (referenced

paper by Kolb et al, 2022) distinguish between physiology-bias and pathway-bias, which are measured using an endogenous and pathway-balanced reference ligand, respectively. In Figure 1D, serotonin appears to be pathway-balanced, as the Gq dissociation and β -arrestin2 recruitment concentration-response curves are near-identical. The authors should state more clearly in the text whether serotonin is pathway-balanced in their assays- and systems (it is not in studies using other assays or systems).

Thank you for this clarification. We have added that 5-HT is pathway-balanced in our system and cited the Kolb et al. 2022 paper.

3. As a further note on the guidelines for reporting of GPCR ligand bias, I would please ask that the authors

- a) include an updated supplementary table of “Experimental parameters critical to the unambiguous description of ligand bias” (a template is provided in Table 3).
- b) Deposit their 5-HT_{2A} Gq dissociation and β -arrestin2 recruitment data in a public database, most relevant being the Biased Signaling Atlas (<https://biasedsignalingatlas.org>).

Thank for you for this comment. We have included Supplementary Table 1 detailing the experimental parameters used for ligand bias determination. We will deposit the data in the Biased Signaling Atlas upon acceptance.

4. Induced fit docking (IFD) is generally considered as a relatively unprecise docking protocol, especially when multiple residues are selected (for Ala mutation in initial docking and subsequent sidechain placement relative ligand). As a minimum, the authors should:

- a) describe in Methods which residues were selected in the induced fit docking.

We have added this information to the methods.

- b) explain why not some or all compounds could be docked directly into the 25-CN-NBOH bound structural template.

A subset of key compounds was chosen for IFD to assess the binding modes and to provide potential insight into the SAR around affinity and ligand bias. Additional IFD experiments were performed on other 25N ligands but not included as these were consistent with the selected examples.

- c) visualize how the docked poses compare to that of 25-CN-NBOH in the experimental structure template.

We have added this visualization to Supplementary Fig 5A, which shows an overlay between 25CN-NBOH with IFD and experimental structure pose.

Minor comments

L32-33: Here we show that 5-HT_{2A} primarily couples to Gq/11 and β -arrestin2 and that prototypical psychedelics do not show a preference for either effector... Given that the couplings have been shown before, and that authors use “confirm” later in the text, please rephrase to:
Here we confirm that 5-HT_{2A} primarily couples to Gq/11 and β -arrestin2 and show that prototypical psychedelics do not show a preference for either effector...

To conserve space, we have re-phrased as “prototypical psychedelics activate both 5-HT_{2A}-Gq/11 and β -arrestin2 signaling...”

L72-73: The 5-HT_{2A} receptor, like other GPCRs, can exhibit functional selectivity or biased agonism, which is the ability to stabilize subsets of its downstream effectors, such as Gq/11 coupling and β -arrestin2 recruitment. This is not a correct definition of biased agonism. Furthermore, biased agonism typically only refers to the ligand whereas the term functional selectivity encompasses both the ligand and system. Authors should use other definition(s). This can be found in the guideline paper or other authoritative reviews.

Thank you for this correction. We have revised the sentence as per Reviewer#1’s suggestion.

L75-77: For example, the G protein-biased μ -opioid receptor (MOR)-selective agonist oliceridine was recently approved in the US and is claimed to have an improved tolerability profile (Gan and Wase, 2020). This claim should never be mentioned without the strong contradictive evidence (e.g., Gillis, A. et al. Sci. Signal. 13, eaaz3140 (2020)). The vast majority of the GPCR community no longer believes that signaling bias underlies the potential therapeutic benefits of Oliceridine. The sentence could also be removed altogether.

We have cited this study as contradictory evidence for oliceridine.

L229-231: This W6.48 residue, sometimes termed the “toggle switch”, has been implicated in GPCR activation and signaling bias and distinct W6.48 rotamer conformations occur in activated and non-activated GPCR structures (Chen et al., 2022; Piekina-Ciesielska et al., 2020; Suomivuori et al., 2020) W6.48 does not undergo a sidechain rotamer shift upon activation in most class A GPCRs but is instead part of a TM6 helix backbone rotation (see e.g., <https://doi.org/10.1038/s41594-021-00674-7>). Thus, this should no longer be presented as a general activation switch, although in individual GPCRs or active states differences can occur. This should be clarified to make a correct statement.

Thank you for this clarification. We have cited this study and clarified general features of TM6 and additional motifs as included below.

Furthermore, did the authors see differences in the rotation of TM6 and W6.48 across different agonist-bound conformations in their MD analysis? If so, please include that information too.

In the current study, we only compared 25CN-NBOH (experimental ligand) with 25N-N1-Nap. Future studies will use MD and mutagenesis experiments to explore how other compounds influence rotation of TM6 and W6.48.

L469-472: For example, influence of the W6.48 toggle switch on the PIF motif and corresponding changes in TM6 orientation were observed in MD studies with β 2-adrenergic receptor biased ligands (Chen et al., 2022), rhodopsin (Zhou et al., 2017), MOR (Piekielna-Ciesielska et al., 2020), and S1PR1 (Xu et al., 2022).

The PIF motif is actually PIW in the majority of class A GPCRs (<https://doi.org/10.1038/s41594-021-00674-7>). This is actually a previous report of the interaction between W6.48 and the previous PIF motif.

Thank you for the suggestion. We have cited this study and clarified in the discussion “PIF/PIW” motifs.

Reviewers' Comments:

Reviewer #1:

Remarks to the Author:

Each of my concerns has been sufficiently addressed.

Reviewer #2:

Remarks to the Author:

Two main weaknesses are still in the revised version of the manuscripts:

1- the authors show that some ligands induce b-arrestin recruitment, but this does not provide any information about signaling downstream b-arrestin , or if there is merely the traditional GPCR-G protein uncoupling mechanisms mediated via b-arrestin recruitment.

2- Behavior models are extremely weak and totally unrelated to the main conclusions of this manuscript

Reviewer #3:

Remarks to the Author:

Authors have incorporated majority of suggestions and comments. My only concern relates to the PK/PD correlation. I may understand authors' argumentation on effects observed in vivo and their CNS-specificity, yet relevance of the effects at tested doses should be confirmed in next step of the compounds development. Given that agonism has been measured in an overexpressed recombinant cell system, possibly overestimating the potency, further confidence in compound activating properties should be confirmed.

The manuscript deserve to be published in Nature Communications.

Reviewer #4:

Remarks to the Author:

The rebuttal and revised manuscript have addressed all comments satisfactory. Thus, I have not further comments and can recommend publication of this highly interesting study in Nature Communications.

REVIEWER COMMENTS

Reviewer #1 (Remarks to the Author):

Each of my concerns has been sufficiently addressed.

Thank you for your careful assessment of our work.

Reviewer #2 (Remarks to the Author):

Two main weaknesses are still in the revised version of the manuscripts:

1- the authors show that some ligands induce b-arrestin recruitment, but this does not provide any information about signaling downstream b-arrestin , or if there is merely the traditional GPCR-G protein uncoupling mechanisms mediated via b-arrestin recruitment.

Thank you for this comment. We have determined that the effect is merely an uncoupling mechanism via β -arrestin recruitment and subsequent internalization. The manuscript now clarifies that the effects produced by the compounds are distinct from “signaling” and we specify exactly what is being measured with respect to β -arrestin throughout the manuscript.

2- Behavior models are extremely weak and totally unrelated to the main conclusions of this manuscript

Although we show efficacy with the β -arrestin biased compounds in a PCP-induced hyperlocomotor activity assay (which is a typical preclinical screening model used to assess whether 5-HT2A antagonists have anti-psychotic potential), we have toned down the discussion of the therapeutic effects with these compounds and include a paragraph in the discussion acknowledging the limitations of our approaches.

Reviewer #3 (Remarks to the Author):

Authors have incorporated majority of suggestions and comments. My only concern relates to the PK/PD correlation. I may understand authors' argumentation on effects observed in vivo and their CNS-specificity, yet relevance of the effects at tested doses should be confirmed in next step of the compounds development. Given that agonism has been measured in an overexpressed recombinant cell system, possibly overestimating the potency, further confidence in compound activating properties should be confirmed.

The manuscript deserve to be published in Nature Communications.

Thank you for this comment. As we detailed in the previous Reviewer Response to Reviewer#1, G protein dissociation BRET assays are not susceptible to overexpressed amplified potency estimates since they are not susceptible to receptor reserve. In fact, many of the affinity values obtained in radioligand binding experiments are on par with the EC50 measurements in the BRET assays, suggesting little overestimation of the potency. That being said, we have included a paragraph in the

discussion on the limitations of our approaches and acknowledge the need to determine optimal PK/PD parameters for proper dosing.

Reviewer #4 (Remarks to the Author):

The rebuttal and revised manuscript have addressed all comments satisfactory. Thus, I have not further comments and can recommend publication of this highly interesting study in Nature Communications.

Thank you for your assessment of our manuscript.